# CRISPR-CasRx-mediated disruption of *Aqp1/Adrb2/Rock1/Rock2* genes reduces intraocular pressure and retinal ganglion cell damage in mice

Mingyu Yao [1,2,5], Zhenhai Zeng[1,2,5], Siheng Li[1,3,5], Zhilin Zou[3,5], Zhongxing Chen[1,2], Xinyi Chen[3], Qingyi Gao [1,2], Guoli Zhao[1,2], Aodong Chen[3], Zheng Li[3], Yiran Wang[1,2], Rui Ning[1,2], Colm McAlinden [1,4], Xingtao Zhou[1,2] ✉ & Jinhai Huang [1,2] ✉

Glaucoma affects approximately 80 million individuals worldwide, a condition for which current treatment options are inadequate. The primary risk factor for glaucoma is elevated intraocular pressure. Intraocular pressure is determined by the balance between the secretion and outflow of aqueous humor. Here we show that using the RNA interference tool CasRx based on shH10 adenovirus-associated virus can reduce the expression of the aqueous humor circulation related genes *Rock1* and *Rock2*, as well as aquaporin 1 and β2 adrenergic receptor in female mice. This significantly reduced intraocular pressure in female mice and provided protection to the retina ganglion cells, ultimately delaying disease progression. In addition, we elucidated the mechanisms by which the knockdown of *Rock1* and *Rock2*, or aquaporin 1 and β2 adrenergic receptor in female mice, reduces the intraocular pressure and secures the retina ganglion cells by single-cell sequencing.

Glaucoma, the second leading cause of irreversible blindness worldwide, is characterized by optic neuropathy and visual field loss[1,2]. It is estimated that there are currently 80 million people suffering from glaucoma, with 74% of cases are the primary open angle glaucoma (POAG) form[3,4]. There is a trend that the number of people with glaucoma is increasing year by year and is expected to increase to 111.8 million by 2040[5]. Elevated intraocular pressure (IOP) is the most important risk factor for glaucoma[6-11]. Currently, the only proven effective and widely accepted treatment for glaucoma is reduction of IOP[12]. While long-term use of IOP-lowering drops is effective for most patients, poor patient compliance and side effects are serious concerns, limiting the current practical effectiveness of these drops[13,14].

Surgical treatments such as minimally invasive glaucoma surgery and trabeculectomy are invasive and associated with many complications. Therefore, there is an urgent need to develop more advanced clinical tools and approaches to offer treatments that are both longer-lasting and safer.

IOP arises as the balance between aqueous humor (AH) production by the ciliary body (CB) and its outflow via pathways including the uveoscleral route and trabecular meshwork (TM)[15,16]. At present, the main treatment approaches focus on these two pathways. Aquaporin 1 (*Aqp1*), a family of water-transporting transmembrane proteins mainly expressed in the CB, is involved in the secretion of AH[17-20]. A previous study has shown that disrupting *Aqp1* using CRISPR-Cas9 reduced IOP

[1]Eye Institute and Department of Ophthalmology, Eye & ENT Hospital, Fudan University; NHC Key laboratory of Myopia and Related Eye Diseases; Key Laboratory of Myopia and Related Eye Diseases, Chinese Academy of Medical Sciences, Shanghai, China. [2]Shanghai Research Center of Ophthalmology and Optometry, Shanghai, China. [3]School of Ophthalmology and Optometry and Eye Hospital, Wenzhou Medical University, Wenzhou, Zhejiang, China. [4]Corneo Plastic Unit & Eye Bank, Queen Victoria Hospital, East Grinstead, UK. [5]These authors contributed equally: Mingyu Yao, Zhenhai Zeng, Siheng Li, and Zhilin Zou. ✉e-mail: xingtaozhou@fudan.edu.cn; jinhaihuang@fudan.edu.cn

by 22% in experimental glaucoma mouse models compared to controls[21]. However, single-gene knockout therapy for *Aqp1* is limited and multiple targets need to be destroyed to extend efficacy. β2 adrenergic receptor (*Adrb2*) blockers are one of the most commonly prescribed drugs for the treatment of glaucoma. Reduction of *Adrb2* expression using siRNA compounds has been shown to reduce IOP by decreasing CB AH production, but this has not been possible to sustain for long periods of time[22]. Moreover, increasing AH outflow is also very important in the treatment of raised IOP. Several studies have also shown that the cytoskeleton regulation related Rho kinase signaling pathway is associated with AH outflow. ROCK family kinases (*Rock1* and *Rock2*) are among the most studied RhoA effectors[23]. The use of Rho kinase and its effector related inhibitors, or siRNA, can significantly affect the morphology of TM cells, effectively increasing the AH outflow coefficient and thus reducing the IOP, and is effective in neuroprotection[23–28]. The aforementioned aspects highlight the appeal of targeting kinases in the ROCK family.

Given that glaucoma is a chronic neurodegenerative disease requiring long-term treatment, gene editing is an attractive approach to provide a relatively permanent therapeutic alteration. The CRISPR/Cas13 system, which belongs to the class 2 type VI CRISPR/Cas tools, is a novel RNA interference tool whose family member, CasRx, can efficiently knock out the target mRNA with high specificity and without significant off-target effects[29,30]. Due to its compact structure and small size, CasRx makes it easier to use AAV vectors for in vivo delivery[30–32]. Compared to other gene-editing systems such as CRISPR/Cas9, the CasRx system only edits RNA without affecting the original DNA

sequence and is a safer gene-silencing method[33]. CasRx for RNA interference has been demonstrated to have great potential applications.

Here, we show that AAV viruses of the shH10 serotype can efficiently transduce the TM and CB for CasRx system delivery. Using two different mouse models of glaucoma, we show that reducing the expression of *Aqp1* and *Adrb2* in the CB or *Rock1* and *Rock2* in the TM, significantly reduces IOP, protects retinal RGC cells and delays disease progression. We also show that sustained, long-term reductions in the expression of these genes are safe and well-tolerated.

## Results

### CasRx achieves knockdown of the *Rock1*, *Rock2*, *Aqp1* and *Adrb2* genes in vitro

To examine the efficiency of CasRx-mediated knockdown of *Rock1*, *Rock2*, *Aqp1*, and *Adrb2*, we designed several 30 nt gRNAs sequences for each gene (Fig. 1a). Two plasmids, including the gRNAs with mCherry expression vector, CasRx-GFP fluorescence vector, were constructed (Fig. 1b). We found that co-transfection of vectors containing CasRx with gRNA 4 that target *Rock1* exon 6 and *Rock2* exon 25, resulted in 81% ± 5% and 81% ± 3% reduction of *Rock1* and *Rock2* mRNA in N2a cells respectively (Fig. 1c, d). Therefore, we constructed Y25 virus vector AAV-EFS-CasRx/U6-*Rock1*-*Rock2* (with gRNAs 4 targeting *Rock1* and *Rock2*) in order to increase the AH outflow in TM cells (see below).

For illustrating editing efficiency of the *Aqp1* and *Adrb2* genes in N2a cells, stable cell lines were constructed due to their low

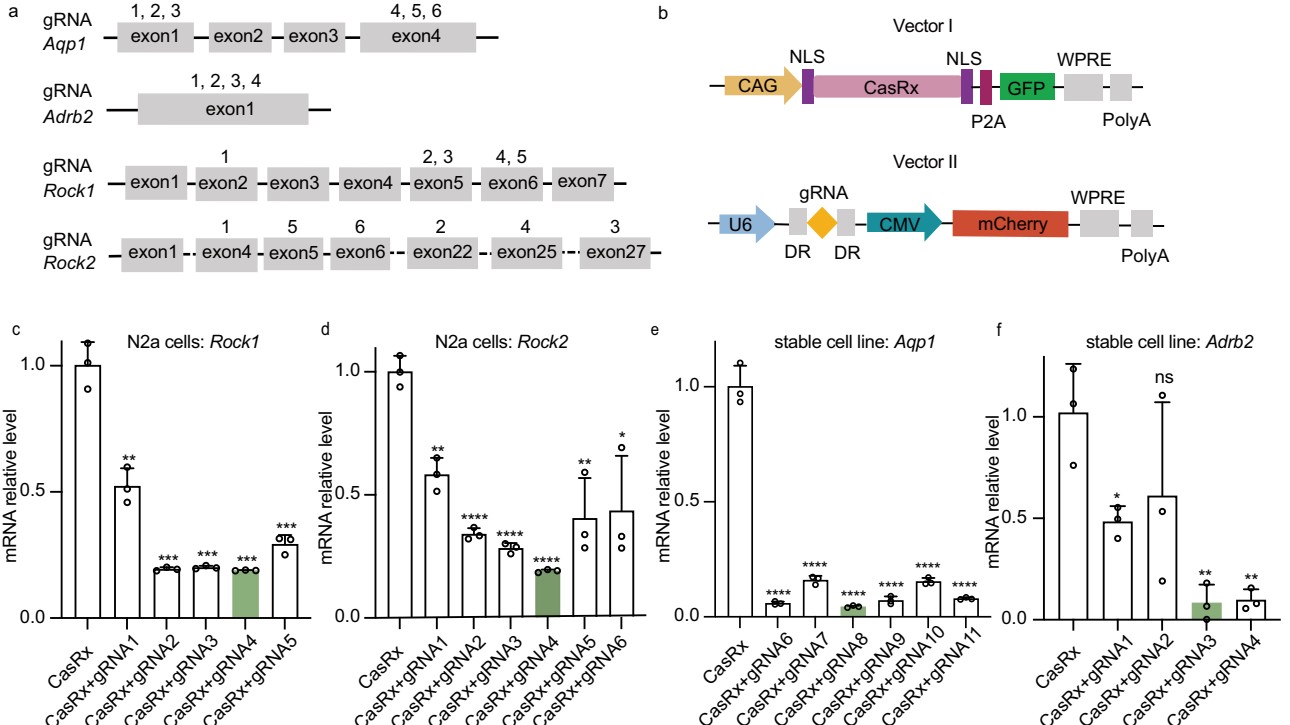

**Fig. 1 | In vitro screening for highly effective and specific gRNAs for target genes. a** The target locations of gRNA in *Aqp1*, *Adrb2*, *Rock1* and *Rock2*. **b** CasRx expression vectors (Vector I) with GFP fluorescence signal and gRNA expression vectors (Vector II) with mCherry separate targeting *Aqp1*, *Adrb2*, *Rock1* and *Rock2* were constructed. **c** Knockdown efficiency of five different gRNAs targeting the *Rock1* gene in N2a cells, $n = 3$. Compared with group CasRx, the gRNA4 showed the highest knockdown efficiency, $P = 0.0001$. **d** Knockdown efficiency of six different gRNAs targeting the *Rock2* gene, $n = 3$. Compared with group CasRx, the gRNA4 showed the highest knockdown efficiency in N2a cells, $P < 0.0001$. **e** The knockdown efficiency of six different gRNAs targeting *Aqp1* gene

was verified in stable transfected N2a monoclonal cell strain No. 6 with high expression of *Aqp1* gene, $n = 3$. Compared with group CasRx, the gRNA3 showed the most potent knockdown efficiency, $P < 0.0001$. **f** Knockdown efficiency of four different gRNAs targeting *Adrb2* gene in No. 16 stable N2a cell lines with high expression of the target gene, $n = 3$. Compared with group CasRx, the gRNA3 showed the strongest knockdown potential, $P = 0.0032$. All values were expressed as mean ± SD, *$P < 0.05$, **$P < 0.01$, ***$P < 0.001$, ****$P < 0.0001$, ns no signafance, two-tailed, unpaired *T*-test. Source data are provided as a Source Data file.

expression. CRISPR/Cas9 technology was employed to enhance the expression of target genes by inserting the CMV promoter and hPGK-puro-polyA sequence before the transcription start site (TSS) of the *Aqp1* and *Adrb2* gene sequences in N2a cells. The sgRNA-Cas9 vector for knock-in and the vector expressing the donor fragment (Fig. S1b) were co-transfected into N2a cells. Following transfection, cells that successfully incorporated the desired genes were subjected to pur-omycin drug killing and flow sorting for screening (Fig. S1a). In the lysates of these cells, *Aqp1* mRNA levels were found to be up to 215-fold higher (Fig. S1c) and *Adrb2* mRNA levels up to 9.85-fold higher (Fig. S1d) compared to wild-type N2a cells. The insertion site of the cultured monoclonal cell line genome was identified using nested PCR and first-generation sequencing (Fig. S2a–f). The No. 6 monoclonal N2a stable cell line, which exhibited high expression of *Aqp1*, and the No. 16 monoclonal N2a stable cell line, which showed high expression of *Adrb2*, were selected. Using established cell lines, we found that co-transfecting CasRx-containing vectors with gRNA3 targeting *Aqp1* exon 1 and *Adrb2* exon 1 resulted in 95% ± 5% reduction in *Aqp1* mRNA and 93% ± 14% reduction in *Adrb2* mRNA in N2a cells, respectively (Fig. 1e, f). To enhance the effect of inhibiting the AH production, we constructed the Y26 virus vector AAV-EFs-CasRx/U6-*Aqp1-Adrb2* (with gRNAs 3 targeting *Aqp1* and *Adrb2*). Additionally, we created the LacZ virus vector AAV-EFS-CasRx/U6-LacZ as a non-targeting control virus for future studies (see below).

### Screening for the effects of AAV infection in mouse eyes and the establishment of two mouse models of ocular hypertension

To screen for the AAV serotypes capable of effectively infecting CB and TM, we intravitreally injected the shH10 serotype viruses, which was known to be effective in infecting CB[21] and AAV2 serotype, with the most commonly used serotype for eye injections. One week after intravitreal virus injection, we observed the expression of the shH10 serotype in the CB (indicated by the arrow) and the TM (indicated by the box) (Fig. 2a, b). The strong GFP signal persisted even 6 months after injection (Fig. 2c). Subsequently, we utilized the shH10 serotype AAV for CasRx system delivery. We also systematically evaluated the ocular safety of intravitreal injection of shH10 virus (Fig. S14 and Fig. S16). Neither shH10 Y25 virus nor shH10 Y26 virus had any effect on corneal endothelium, angular structure, retina or visual function.

We also examined two mouse models of ocular hypertension that mimic glaucoma. One of these models, known as the magnetic microbead occlusion model, was first described in a publication in 2016[34] and has since been widely referenced in glaucoma and optic nerve injury studies[35–37]. In this model, we injected micromagnetic beads into the anterior chamber of the eye to induce ocular hypertension. We measured IOP during the daytime (9:00–12:00) and nighttime (20:00–23:00) every week (Fig. 2d). We observed that the IOP of the mice injected with magnetic beads ranged from 12.89 ± 0.31 mmHg at the first week to 20.85 ± 0.50 mmHg at the 6th week. The nocturnal IOP followed a similar trend (Fig. 2e). Survival retina ganglion cells (RGCs) in the central, middle and peripheral regions of retinas at 6 weeks were quantified by Rbpms immunostaining (Fig. 2d). Statistical analysis revealed that the mice modeled with magnetic beads exhibited a significant, 16.9% reduction, in Rbpms-positive RGCs compared to wild-type mice (Fig. 2f, g).

Animal models of glucocorticoid-induced ocular hypertension have been shown to closely resemble clinical POAG, exhibiting similar TM lesions[38–42]. In 2020, Maddineni et al. optimized and proposed a model for inducing high IOP simulated open-angle glaucoma in mice via weekly injections of dexamethasone acetate suspension through the conjunctival fornix[43]. For this experiment, 3 month-old female C57BL/6 J mice were used as model subjects. After measuring the baseline IOP, 10 µl of dexamethasone acetate suspension was injected weekly through the conjunctival fornix, and the daytime and nighttime IOPs were measured (Fig. 2d). The results showed that the IOP of mice in the continuous hormone injection group was higher than that of the non-injection WT group, indicating the successful establishment of this chronic high IOP model (Fig. 2h). Rbpms-positive cell counting indicated that the mice modeled with DEX exhibited a 12.2% reduction in RGCs compared to wild-type mice at 10th week (Fig. 2l, j).

### CasRx leads to efficient disruption of the *Rock1*, *Rock2*, *Aqp1* and *Adrb2* transcripts in mice

Ten weeks after viral injection, samples were collected and the infection effect was assessed by staining the flag tag protein that was attached to the virus (Fig. S3a, b). In the magnetic beads model mice and DEX-induced model mice, after injection of shH10 Y25/Y26 virus and shH10 LacZ virus, co-staining with DAPI revealed the presence of the red fluorescent signal of the flag protein in the TM region and CB region (as indicated in the box).

We evaluated target gene expression in the eyes of mice using fluorescence as a semi-quantitative method. The results indicated that compared to the LacZ virus group, the injection of Y25 virus in mice injected with magnetic beads led to a reduction in the expression levels of *Rock1* gene and *Rock2* gene in the TM by 44.1% ($P = 0.0020$) and 63.0% ($P = 0.0069$) respectively. There were no significant differences in gene expression levels between the LacZ group, WT group, and saline injection group (Fig. 3c–e). Similar results were observed in the TM of hormone-induced model mice injected with shH10 Y25 virus into the vitreous (Fig. 3f–h). In the established high IOP mice modeled with magnetic beads, the expression levels of the *Aqp1* gene and *Adrb2* gene in the CB were significantly decreased by 43.7% ($P = 0.0192$) and 54.0% ($P = 0.0121$), respectively, in the shH10 Y26 virus group compared to the LacZ virus group. There was no significant difference between the WT group and the model mice injected with normal saline or the LacZ virus group (Fig. 3i–k). The shH10 Y26 virus also exhibited a similar knockdown effect on the CB of DEX-induced model mice (Fig. 3l–n). Furthermore, we investigated viral infection at various organ sites across the body. No fluorescent signal of the red flag co-stained with DAPI was detected outside the eye (Figs. S4–5). Otherwise, H&E staining results demonstrated that there were no observable pathological changes in multiple organs (Fig. S15). These experimental findings demonstrate that the intravitreal injection of the shH10 virus carrying the CasRx-gRNA system into the mouse eye effectively induces RNA interference in the TM tissue and CB, resulting in the targeted gene transcript destruction without impacting other organs, thus ensuring systemic safety.

### Gene knockdown mediated by the CasRx editing system effectively reduces IOP and RGCs damage in two glaucoma models

Next, we evaluated the therapeutic effect on glaucoma using the CasRx system to interfere with the expression of the *Rock1* and *Rock2* genes, as well as the *Aqp1* and *Adrb2* genes, on two simulated glaucoma model mice (Fig. 3a). As depicted in Fig. 3a, the mice were subjected to the therapeutic intervention 3 days after the establishment of the ocular hypertension model. The daytime and nighttime IOP were consistently monitored, and the RGCs were quantified after 10 weeks. In the group that was injected with normal saline and LacZ virus after the magnetic bead modeling, the IOP of the mice significantly increased from the first week and remained stable until the sixth week. When compared with the mice treated with LacZ virus, the group treated with shH10 Y25 virus showed a significant decrease in IOP, with an average reduction of 5.78 ± 0.76 mmHg (Fig. 4a). This result demonstrates the high effectiveness of shH10 Y25 viruses in the clinical treatment of glaucoma. Similarly, the group treated with the shH10 Y25 viruses in the hormone-induced model also exhibited a decrease in IOP (Fig. 4b). Furthermore, the data obtained after 10 weeks of virus injection revealed a significant reduction in the number of Rbpms positive RGCs in the center, mid-peripheral, and peripheral regions of the group

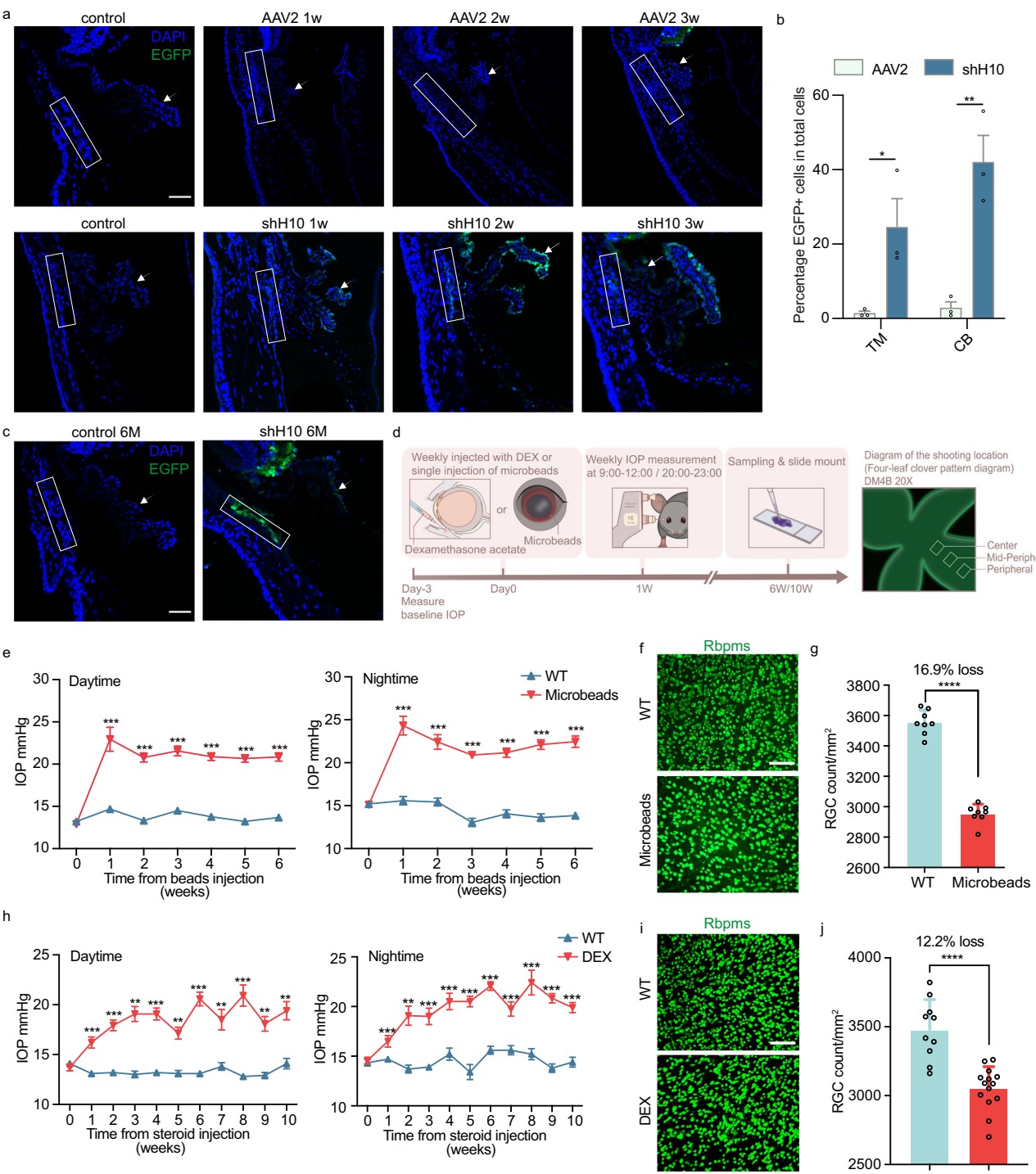

**Fig. 2 | Evaluation of AAV virus infection and establishment of two intraocular hypertension models in mice. a** Intraocular infection of C57BL/ 6 J female mice after intravitreal injection of AAV2-EFS-EGFP or shH10-EFS-EGFP. The eyeball was taken at 1 week, 2 weeks and 3 weeks after injection. The arrow indicated the ciliary body (CB) and the box indicated the trabecular meshwork (TM), $n = 3$. Scale bars,100 μm. **b** Two weeks after virus injection, the percentage of EGFP+ cells in the TM ($P = 0.0389$) and CB ($P = 0.0058$) area in total cells, $n = 3$. Data were expressed as mean ± SEM. **c** Six months after intravitreal injection of virus, the shH10-EFS-EGFP was still expressed in CB and TM, $n = 2$, Scale bars, 100 μm. **d** The establishment and evaluation process of two intraocular hypertension mouse models. RGCs of Rbpms + were counted from 6–12 fields in each retina at 6 weeks after microbeads injection or 10 weeks after DEX modeling. **e** Compared with wild-type (WT) mice of the same age, the daytime and nightime IOP of the mice injected with magnetic beads was $7.154 ± 0.5457$ mmHg higher than that of the WT mice at the 6th week, and the nightime IOP of the mice was $8.615 ± 0.7634$ mmHg higher than that of the WT mice. $n = 18$, all values were expressed as mean ± SEM. **f, g** RGC count result, $P < 0.0001$, $n = 8$, and the data were expressed as mean ± SD. Scale bars, 100 μm. **h** The changes of daytime and nightime IOP of DEX-induced mice ($n = 14$) in 10 weeks compared with WT mice ($n = 10$) of the same age, all values were expressed as mean ± SEM. **i, j** The result of RGC counting. WT, $n = 10$, DEX, $n = 14$, $P < 0.0001$. The data were expressed as mean ± SD. All data were analyzed by two-tailed, unpaired T-test without adjustments for multiple comparisons, *$P < 0.05$, **$P < 0.01$, ***$P < 0.001$, ****$P < 0.0001$. Source data are provided as a Source Data file.

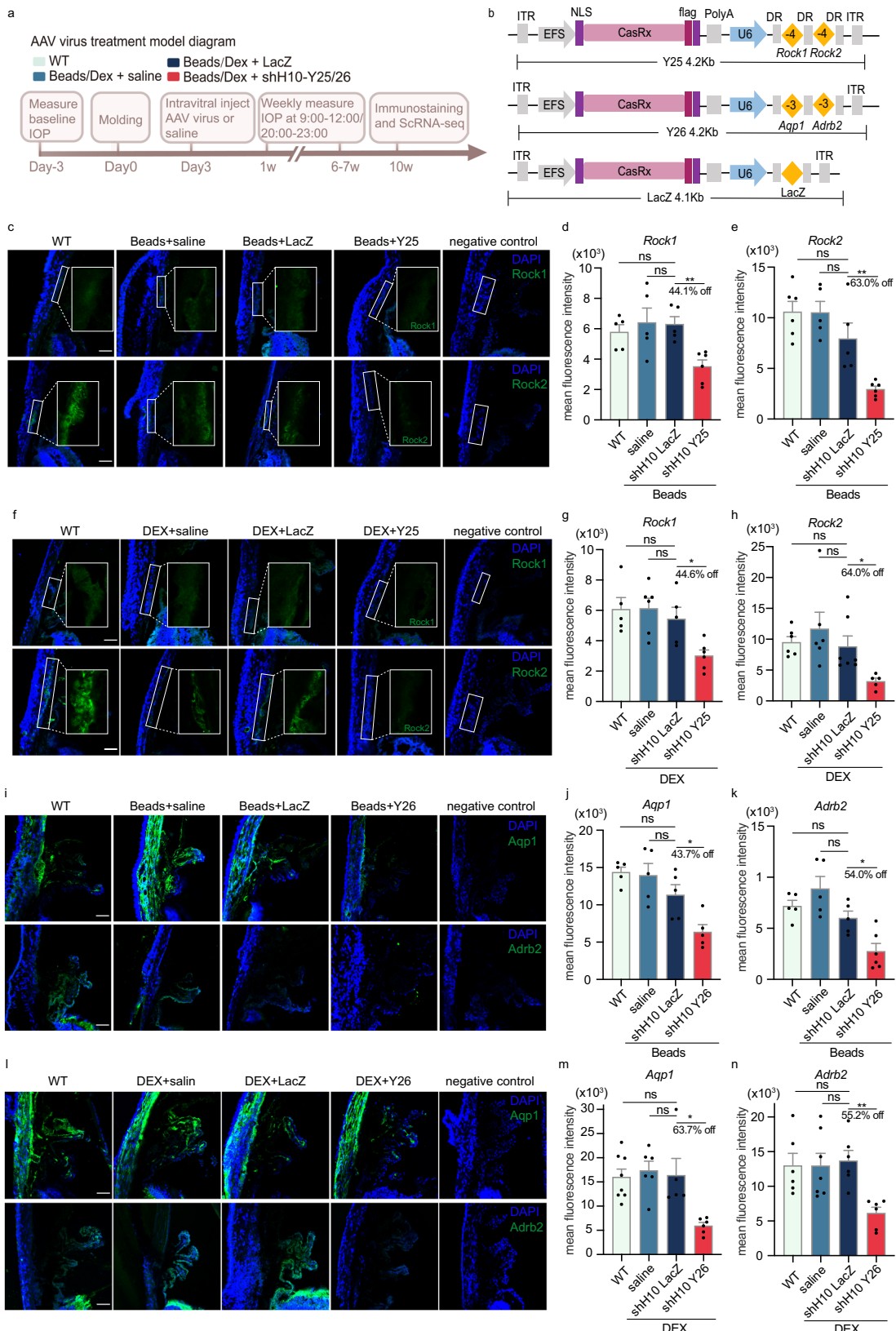

injected with saline and shH10 LacZ, compared to that of WT mice (Fig. 4c–f). Notably, the loss of RGCs was most pronounced in the peripheral retina. Importantly, the injection of the therapeutic viruses shH10 Y25 demonstrated a significant reduction in the death rate of RGCs, indicating that CasRx's interference with the expression of the *Rock1* and *Rock2* genes in TM not only reduced the IOP, but also

mitigated the loss of RGCs. In addition, we investigated the impact of suppressing the *Aqp1* and *Adrb2* genes on glaucoma treatment. Using two well-established models of high IOP, we observed that the group treated with the shH10 Y26 virus showed a significant decrease in both daytime and nighttime IOP in mice (Fig. 4g, h). Furthermore, this treatment demonstrated a positive effect in protecting ganglion cells

**Fig. 3 | CasRx disrupted the target gene transcript in mice. a** Experimental strategy of magnetic beads-induced and dexamethasone acetate induced intraocular hypertension mouse modes. **b** Three kinds of AAV virus vector encoding CasRx and gRNAs with the highest knockout efficiency. **c–e** Compared with non-target LacZ virus (*n* = 5), the CasRx system delivered by shH10 Y25 (AAV-EFS-CasRx/U6-*Rock1*-*Rock2*) virus (*n* = 6), significantly disrupted the expression levels of *Rock1* and *Rock2* genes in the TM of the mouse model injected with microbeads (*P* = 0.0020, *P* = 0.0069). The figure (**d**, **e**) were the statistical results of *Rock1* and *Rock2* expression levels after semi-quantification by immunofluorescence. *Rock1*/ *Rock2* saline, *n* = 5; *Rock1* WT, *n* = 5; *Rock2* WT, *n* = 6. **f–h** The delivery of CasRx system by shH10 Y25 into the eye can significantly reduce the expression levels of *Rock1* and *Rock2* genes in trabecular meshwork of DEX-induced mice (*P* = 0.0147, *P* = 0.0218). The g and h images were fluorescence quantization results. *Rock1* WT /saline/LacZ/Y25, *n* = 5, 6, 5, 6; *Rock2* WT /saline/LacZ/Y25, *n* = 6, 6, 7, 5. **i–k** In

contrast to non-target LacZ virus, shH10 Y26 (AAV-EFS-CasRx/U6-*Aqp1*-*Adrb2*) virus delivery of CasRx system into the eye can significantly damage *Aqp1* and *Adrb2* gene transcripts in the ciliary body of magnetic bead model mice (*P* = 0.0192, *P* = 0.0121). The expression levels of *Aqp1* and *Adrb2* were determined by semi-quantitative immunofluorescence analysis in (**i**) and (**j**) charts. *Adrb2* Y26, *n* = 6; other group, *n* = 5. **l–n** shH10 Y26 (AAV-EFs-CasRx/U6-*Aqp1*-*Adrb2*) virus delivery of CasRx system into the eye can significantly interfere with *Aqp1* and *Adrb2* gene expression in the ciliary body of DEX-induced model mice (*P* = 0.0101, *P* = 0.0013). The expression levels of *Aqp1* and *Adrb2* in m-chart and n-chart were determined by semi-quantitative immunofluorescence analysis. *Aqp1* WT /saline/LacZ/Y26, *n* = 8, 6, 5, 6; *Adrb2* WT /saline/LacZ/Y26, *n* = 6, 7, 6, 6. The box shows TM. In all figures, scale bars, 100 µm. All values were expressed as mean ± SEM, *$P < 0.05$, **$P < 0.01$, ***$P < 0.001$, ****$P < 0.0001$, ns, no significance. two-tailed, unpaired *T*-test; Source data are provided as a Source Data file.

(Fig. 4I, j). We further compared the IOP reduction effects of shH10 Y25 virus, shH10 Y26 virus and Latanoprost on different models of mice within 6 weeks (Fig. S17).

## ScRNA-seq of the mouse CB and contiguous tissue and identification of cell types

In order to investigate the mechanism by which shH10 Y25 or shH10 Y26 reduces the IOP, the mouse CB and contiguous tissue was isolated from shH10 Y25, shH10 Y26 and control ocular hypertension model mice. The droplet-based scRNA-seq was performed, unsupervised clustering on cells for the shH10 Y25, shH10 Y26 and control ocular hypertension model mice using t-distributed stochastic neighbor embedding (t-SNE) revealed 16 clusters with distinct gene expression signatures (Fig. 5a, b; Fig. S6a–c; Fig. S11a, b). We classified these clusters into discrete subpopulations, including rods, RPE, RGC, U-Fibroblasts, S-Fibroblasts, NK cells, Muller cells (MG), rM Macrophages, Cones, Neutrophils, ciliary epithelium cells (CE), B cells, Melanocytes, and Schwamm cells according to the expression of specific markers (Fig. 5c). The specific marker genes are highly cell type specific in distinct subpopulations.

## Inhibition of the *Rock1* and *Rock2* reduces the IOP by inhibiting inflammatory responses

We found that after CRISPR/CasRx knockdowns of *Rock1* and *Rock1* gene expression, the number of N/K T-cell and neutrophils increased significantly compared to the control group (Fig. 5d). The heatmap visualization of gene expression differences between Y25 and saline displayed the top 20 differentially expressed genes (DEGs) (Fig. S9). The differential gene volcano map showed that a large number of genes were upregulated and down-regulated after interference with *Rock1* and *Rock2* expression (Fig. S8a). Among them, we found that these differential genes are enriched in the "antigen processing and presentation of endogenous antigen", "cell killing", and "antigen processing and presentation of endogenous peptide antigen via MHC class", which were upregulated in cells of the Y25 sample after Rock1 and *Rock1* were knocked down. (Fig. S8b). These indicated that the inhibition of *Rock1* and *Rock2* reduces the IOP which may be related to immunological processes. Inflammation is the immune system's response to harmful stimuli. To gain further insight of the functional roles of Y25, we used the GSEA approach to normalized datasets using hallmark pathways gene sets from MSigDB[44]. Normalized enrichment scores (NES) of pathways were plotted. Pathways related to "Tnf-α signaling via NF-κB", "Kras signaling up", "apical junction", and "inflammatory response" were upregulated by sh10 Y25(Fig. S7a–f). It is suggested that the mechanism of inhibiting *Rock1* and *Rock2* to reduce the IOP may be regulated through the aforementioned signaling pathways. VlnPlot of each cell type showed that N/K T-cell and neutrophils contributed the most to the regulation of these signaling pathways. Neutrophils and T cells play a role in eliminating

dysregulated synapses of retinal ganglion cells (RGCs), phagocytosing dead cells and debris, as well as presenting antigens. T-cell-mediated immune responses can initially limit neurodegeneration[45], while recruitment of T cells facilitates early communication between the immune system and harmful stimuli-induced cell debris. This mechanism prevents disruption of immune regulation caused by persistent harmful stimuli, thereby reducing secondary degeneration of retinal ganglion cells (RGCs), which is referred to as "protective immunity"[46–49]. In chronic neurological disability diseases, distinct subpopulations of neutrophils play a pivotal role in promoting RGCs survival and facilitating axon regeneration[50]. These results suggest that the inflammatory response, in which N/K T-cell and neutrophils participate, may play an important role in glaucoma. (Fig. S7g–j).

## Identification of trabecular cell subtypes and their role in regulating AH outflow

Prior analyses of trabecular cells and their role in the AH outflow mechanism are still unclear due to the anatomic structure of trabecular tissue and the limited number of cells to identify and tune the action of the AH outflow. Using single-cell sequencing, we found that there are two important types of fibroid cells in trabecular tissue, S-Fib and U-Fib. U-Fib cells expressed Serpine2 and S-Fib cells expressed slc4a4. The single-cell regulatory network inference and clustering (SCENIC) results show that S-Fib cells were mainly driven by five transcription factors (TFs): *Elk1*, *Twist1*, *Pitx1*, *Pou5f1*, and *Klf1*. U-Fib cells were mainly driven by *Foxd1*, *Foxd2*, *Klf5*, *Pitx3*, and *Twist1* (Fig. 5e, f). Although the two subpopulations are similar, there are significant differences in the expression of certain genes. *Edn3* is expressed only in U-Fib cells and almost not in S-Fib cells (Fig. 5g). ET-3 is a potent vasoconstrictor, and is the robust agonist for endothelin receptor type B (ETB) which expressed on the surface of trabecular cells. In contrast, *Mmp2* and *Mmp3*, which were highly expressed in S-Fib cells, is hardly expressed in U-Fib cells (Fig. 5g, Fig. S10). *Mmp3* contributes to the degradation and remodeling of extracellular matrix (ECM), and promotes the patency of AH outflow channels and the reduction of IOP. The gene set enrichment (GSE) analysis based on Gene Ontology (GO) and Kyoto Encyclopedia of Genes and Genomes (KEGG) also reflected the obvious difference between the two kinds of cells in trabecular tissue. The differential genes upregulated in S-Fib cells relative to U-Fib were mainly related to extracellular matrix deposition in trabecular tissue. For example, collagen fibril organization, extracellular matrix organization, extracellular structure organization, etc. (Fig. 5h). The differential genes upregulated in S-Fib cells relative to U-Fib were mainly concentrated in mesenchyme development, such as neural crest cell development and stem cell development (Fig. 5i). This suggests that the two groups of cells play different roles in regulating the outflow of aqueous oxide. U-Fib cells may regulate the outflow of aqueous by secreting cytokines related to extracellular matrix deposition and morphology of S-Fib cells in the trabecular tissue.

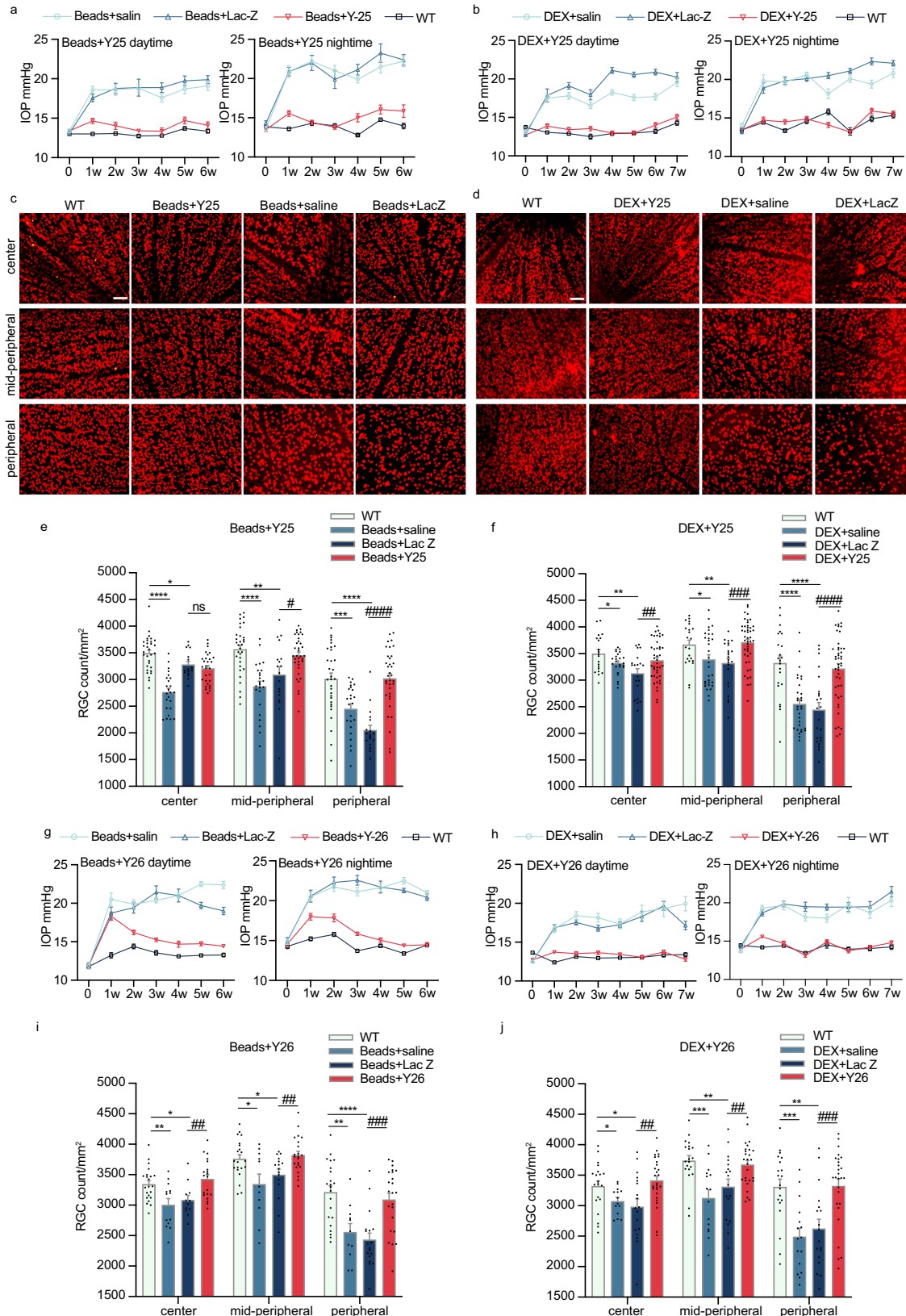

## The mechanism of knocking down the expression of *Aqp1* and *Adrb2* to reduce the IOP

Due to the complexity of the anatomy and the presence of multi-cellular components, the genetic and molecular events underlying the physiological and pathological state of the CB are currently unknown. By using CasRx to knock down the *Aqp1* and *Adrb2* on ciliary cells, we found that it can effectively inhibit the increase of the IOP (Fig. 4I, j). The droplet-based scRNA-seq was performed on cells from the Y26 and control ocular hypertension in model mice. The t-SNE maps revealed 16 clusters with distinct gene expression signatures (Fig. S11a, b; Fig. 5c, Fig. S6a, c). The differential gene volcano map revealed that lower IOP by inhibition of the expression of *Aqp1* and *Adrb2* may

**Fig. 4 | Gene disruption mediated by CasRx editing system effectively reduced IOP and RGC damage in two types of glaucoma models. a, b** IOP changed during day and night in each group after monocular magnetic beads modeling or DEX modeling and injection of shH10 Y25 virus. WT, $n = 20$; Beads+saline $n = 10$; Beads +LacZ $n = 8$; Beads+Y25 $n = 20$. DEX+saline $n = 20$, DEX+LacZ $n = 18$, DEX + Y25 $n = 40$. **c–f** The retinas of microbeads model mice or Dex-induced model mice treated with shH10 Y25 virus were taken 10 weeks after modeling, and the RGC counting results of Rbpms+ were compared among all groups. Each retina takes 6–12 visual fields, including center, mid-peripheral and peripheral positions, Scale bars, 100 μm. **c** and **e** RGC count results of three retinal regions in each microbeads model group. In peripheral, Y25 compared with LacZ, $P < 0.0001$. WT $n = 8$, Beads + saline $n = 6$, Beads + LacZ $n = 5$, Beads+Y25 $n = 8$. **d** and **f** RGC count results of three retinal rings in each Dex-induced model group. In peripheral, Y25 compared with LacZ, $P < 0.0001$. WT $n = 6$, DEX+saline $n = 10$, DEX+LacZ $n = 6$, DEX + Y25 $n = 11$. **g**, **h** Diurnal and nocturnal IOP curves of mice in each group monocular

microbeads modeling or DEX modeling and injection of shH10 Y26 (AAV-EFs-CasRx/U6-*Aqp1-Adrb2*) virus. WT group $n = 18$, Beads+saline group $n = 8$, Beads +LacZ group $n = 7$, Beads+Y26 group $n = 19$. WT $n = 20$, DEX+saline $n = 18$, DEX +LacZ $n = 20$, DEX + Y25 $n = 34$. **i** RGC counting results of center, mid-peripheral and peripheral retinal rings were obtained in microbeads model mice treated with shH10 Y26 virus. In peripheral, Y26 compared with LacZ, $P = 0.0001$. WT $n = 6$, Beads+saline $n = 5$, Beads+LacZ $n = 5$, Beads+Y26 $n = 6$. **j** RGC counting results of three retinal rings were obtained in Dex-induced model mice treated with shH10 Y26 virus. In peripheral, Y26 compared with LacZ, $P = 0.0006$. WT $n = 5$, DEX+saline $n = 4$, DEX+LacZ $n = 5$, DEX + Y26 $n = 7$. All values were expressed as mean ± SEM. The differences of IOP between groups were tested by one-way ANOVA. Other data were statistically analyzed by two-tailed, unpaired $T$-test. Compared with WT, *$P < 0.05$, **$P < 0.01$, ***$P < 0.001$, ****$P < 0.0001$; Compared with LacZ, #$P < 0.05$, ###$P < 0.01$, ####$P < 0.001$.

regulate many related gene expressions (Fig. S12a). The GSE analysis based on GO showed that the differential genes upregulated in cells from Y26 relative to cells from controls were mainly related to immune system process, such as "neutrophil chemotaxis", "neutrophil migration", "myeloid leukocyte migration" (Fig. S12b). Genes related to the production of aqueous solution, such as MYC Targets, PI3K AKT MOTOR SIGNALING, P53 pathway and other gene sets, were down-regulated (Fig. S11c–g). CasRx-mediated knockdown of *Aqp1* and *Adrb2* in the CB can inhibit the expression of apoptosis-related genes (Fig. S11h). This suggests that CasRx-mediated knockdown of the *Aqp1* and *Adrb2* in the CB may have a protective effect against apoptosis. According to the results of SCENIC, cells from the Y26 sample were driven by TFs such as *Atf4*, *Rarb*, *Otx2*, *Bhlhe41* drive (Fig. S11i), while cells from saline control sample were driven by *Thrb*, *Meis1*, *Pbx3*, and *Sp4* (Fig. S11j). The downstream genes regulated by the above TFs are associated with the production of AH, resistance to hypoxia induction, and relaxation of small blood vessels, suggesting that inhibition of *Aqp1* and *Adrb2* expression may affect other cells, such as RGC cells, by regulating the secretion of AH by the CB, and the synthesis of secretion-related regulatory cytokines to salvage the damage caused by elevated IOP.

## Discussion

Glaucoma, an irreversible blinding eye disease, is strongly related to uncontrolled IOP, leading to visual loss[1,2,6–12]. This study presents a novel approach to effectively reduce the IOP using targeted gene combination therapy with the CRISPR-CasRx system. IOP is determined by the balance between aqueous humor secretion and outflow, and there is a homeostasis mechanism responsible for IOP-regulation in the TM/SC outflow pathway[51,52]. Studies have shown that there is a compensatory mechanism involved in maintaining intraocular pressure homeostasis[53–59]. Our strategy involves targeting two specific regions: the TM responsible for AH outflow, and the CB responsible for AH production. The utilization of this approach circumvents the problem of compensatory AH recovery that emerges when focusing on a solitary gene. Moreover, the CRISPR-CasRx system specifically edits RNA without altering the original gene sequence, ensuring higher safety[29,30,33]. It eliminates the need for long-term self-administration and addresses patient compliance concerns. If successfully translated to humans, this innovative treatment approach could greatly benefit glaucoma patients.

Following injection of two of the commonly used AAV serotypes, we observed that shH10 AAV exhibited a more pronounced infection effect on the TM and CB. Utilizing the shH10 AAV for delivering the CRISPR-CasRx system enables efficient target gene knockdown, and a single injection can sustain this effect for a minimum of 10 weeks. Additionally, intravitreal injection demonstrated satisfactory safety profiles, suggesting the potential for clinical translation. To evaluate the efficacy of our treatment, we conducted experiments using two

ocular hypertension mouse models that mimic glaucoma. We observed a reduction in the expression of AH outflow-related genes *Rock1* and *Rock2* in the TM, as well as a decrease in the expression of the AH production-related genes *Aqp1* and *Adrb2* in the CB. Both methods demonstrated a significant and long-term reduction in IOP and provided protection to RGCs cells. Since AAV expression begins 1 week after injection in vivo[60,61], further studies are needed to assess the effect of viral injection at different stages of glaucoma on the course of the disease.However, it remains to be determined whether this approach would have an effect on the AH balance and reduce the IOP in glaucomatous primate or human eyes. Further optimization is required, including the use of glaucoma models in large animals and non-human primates, to assess the efficacy of the intervention and potential side effects, prior to initiating clinical studies.

Neutrophils are important immune cells involved in many inflammatory and immune responses. In glaucoma, neutrophils act to promote the inflammatory response in the eye, aggravating optic nerve damage. Neutrophils can cause an inflammatory response in the eye by releasing substances such as inflammatory mediators and oxygen free radicals, resulting in increased IOP and damage to the optic nerve. In addition, neutrophils can also promote angiogenesis and exudation, aggravating the inflammatory response and injury of the eye. In patients with glaucoma, the number and activity of neutrophils are significantly increased, suggesting that they are involved in the occurrence and development of glaucoma. Another study found that the IOP and optic nerve damage can be reduced by inhibiting neutrophil activity, thus effectively treating glaucoma. In conclusion, the role of neutrophils in glaucoma is to promote an inflammatory response in the eye and to aggravate optic nerve injury. By inhibiting the activity of neutrophils, the progression of glaucoma may be slowed.

It was found that Kras activation protects the RGCs by activating the *Raf/MEK/ERK* signaling pathway. This signaling pathway promotes the survival and proliferation of RGCs, and it is able to resist the damage of RGCs. In addition, *Kras* activation also promotes the synaptic formation and maintenance of RGCs, thereby improving retinal visual function. Specifically, one study has shown that by activating *Kras*, RGC survival and growth can be enhanced, and RGC death can be reduced. Another study found that by activating *Kras*, synaptic formation and maintenance of RGCs can be facilitated, thereby improving retinal visual function. In conclusion, Kras activation can have a protective effect on RGCs, which provides new ideas and methods for possible treatments of glaucoma and other RGC injury-related diseases[62]. Inhibition of the expression and activity of *Rock1* and *Rock1* can promote the expression and activity of the *Kras* gene[63], thus possibly resulting in a positive impact on the treatment of glaucoma.

Increased IOP is usually caused by increased AH production or redcued AH outflow, with the latter playing a greater role. The increase

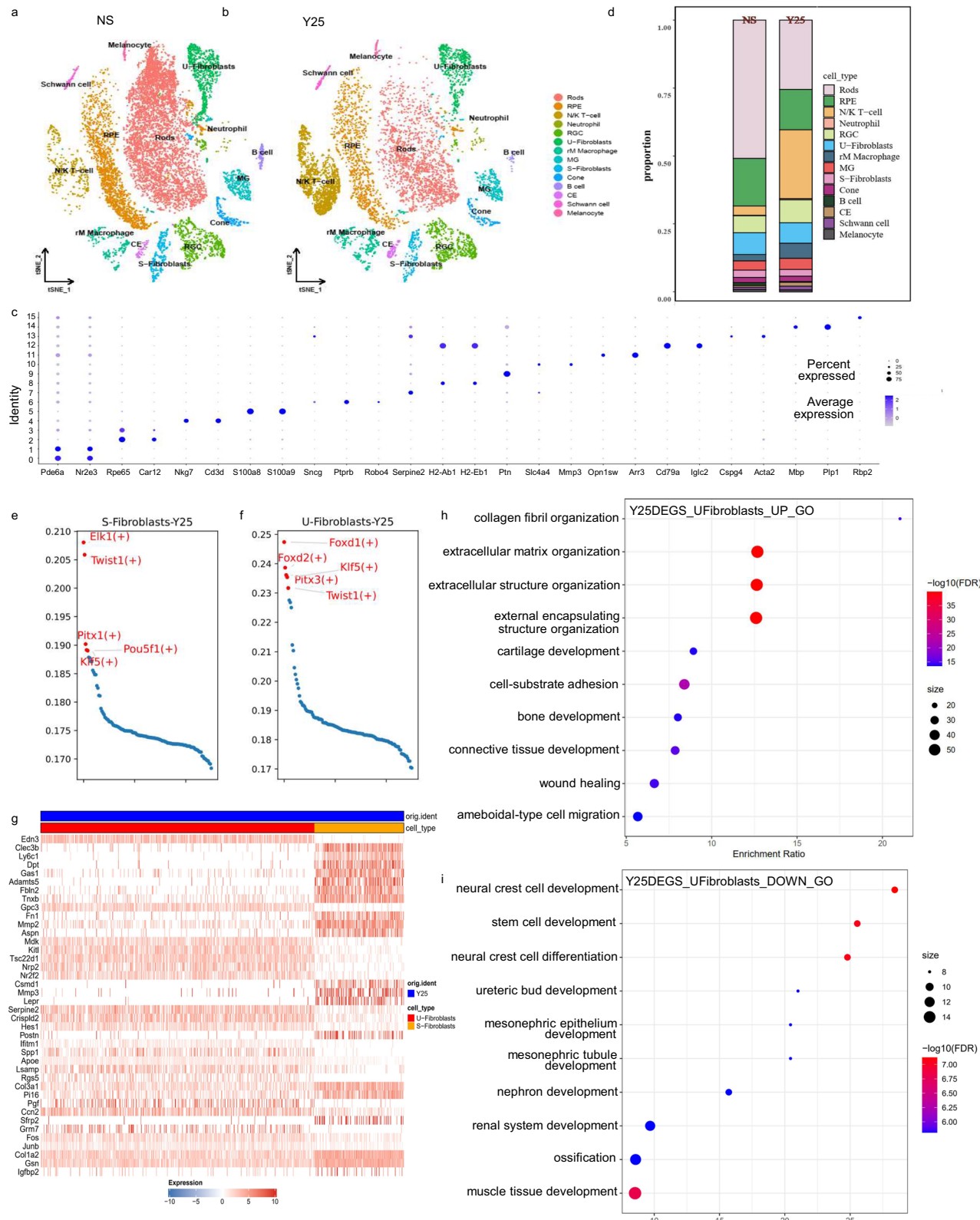

**Fig. 5 | ScRNA-seq of the mouse CB and contiguous tissue from shH10 Y25 and identification of cell types. a** The t-SNE plots showing cell type as determined by marker gene expression from shH10 Y25. **b** The *t*-SNE plots showing cell type as determined by marker gene expression from saline control. **c** Heatmap of marker genes identified through differential expression analysis with respect to clusters. **d** Percentage of cell types of shH10 Y25 group or saline control group. **e**, **f** The single-cell regulatory network inference and clustering (SCENIC) results of S-Fib and U-Fib from shH10 Y25. **g** The heatmap visualization of gene expression differences between S-Fib and U-Fib from shH10 Y25. **h**, **i** The gene set enrichment (GSE) analysis based on Gene Ontology (GO reflected the obvious difference between S-Fib and U-Fib from shH10 Y25).

of extracellular matrix deposition in trabecular tissues is closely related to the obstruction of aqueous outflow channels[64]. The increase of extracellular matrix may be caused by the activation of inflammatory pathways. However, the inflammatory pathway can only occur in vivo. Our data found that compared with saline, Y25 was differentially enriched in signaling pathways such as extracellular matrix generation and collagen generation, suggesting that CasRx knocked down *Rock1* and *Rock2* was remodeled through ECM mediated by inflammation. The phagocytosis of trabecular cells is mediated by inflammatory signaling pathways. Gamma interferons have also been shown to block trabecular phagocytosis. Our data showed that the expression of many genes in Y25 and saline was upregulated and down-regulated, and genes related to inflammatory signaling pathways such as gamma interferon, alpha interferon, and inflammatory response were upregulated in the Y25 group, suggesting that CasRx may reshape trabecular cells by blocking the phagocytotic effect of trabecular tissue by up-regulating gamma interferon related to inflammation. This regulates the outflow AH leading to reduced IOP.

Overall, we demonstrated that CasRx system effectively and specifically reducing the expression of AH outflow-related genes *Rock1* and *Rock2* in the TM, as well as the expression of AH production-related genes *Aqp1* and *Adrb2* in the CB. Moreover, our data showed that a single injection of AAV with CasRx-crRNA can significantly reduce the IOP for at least 6 weeks and provide protection to the RGCs, indicating its potential for long-term control of IOP.

## Methods

### Plasmid construction

The CAG-CasRx-EGFP plasmid and U6-gRNA-mCherry cloning backbone were used. We synthesized gRNAs as single-stranded DNA oligos (all gRNA sequences are provided in Table S1). The gRNA oligos were annealed and cloned under the U6 promoter using the Eco31I enzyme (Thermo Scientific, Massachusetts, USA, FD0294) in the gRNA expression vector for the CasRx system.

The mouse *Aqp1* genomic sequence (NC_000072.7) and *Adrb2* genomic sequence (NC_000084.7) were obtained from National Center for Biotechnology Information. Compatible 20-bp SpCas9 sgRNAs at the sequence position 50 bp before the target gene transcription start site were identified and ranked using DeepSpCas9 according to Kim et al.[65]. Selected sgRNAs were synthesized as oligonucleotides and cloned into the HP180-U6-sgRNA-CMV-Cas9-CMV-EGFP plasmid using the BpiI enzyme (Thermo Scientific, Massachusetts, USA, FD1014). To construct the donor plasmids, 800 bp long sequences on both sides of the Cas9 incision site were cloned as homologous arms using the mouse genome as a template. Then, the fragment hPGK-PuroR-BGH polyA and CMV promotor were cloned into a backbone digested by KpnI (Thermo Scientific, Massachusetts, USA, FD0524) and EcoRI (Thermo Scientific, Massachusetts, USA, FD0247) with the homologous arms by Gibson assembly[66,67].

### AAV production

All AAV used in this study were packaged by PackGene Biotech (Guangzhou, China). AAV2 serotype and recombinant ShH10 serotype viruses encoding an EGFP tag driven by the EFS promoter were used to test the transduction efficiency. The viral titers were 1E + 13GC/ml for the fluorescent viruses. For the AAV plasmid construction, the original gRNA of the AAV-EFS-CasRx plasmid was replaced by the screened gRNAs of the target genes. The AAV-EFS-CasRx/U6-*Rock1-Rock2*, AAV-EFs-CasRx/U6-*Aqp1-Adrb2*, or AAV-EFS-CasRx/U6-LacZ plasmids were constructed and sequenced before being packed into the shH10 vehicle for the following studies. The viral titers were 2E + 13GC/ml. Virus aliquots were separated into small volumes and stored at −80 °C to avoid repeated freeze thaw cycles.

### N2a culture and transfection

The N2a cell line was purchased from the cell bank of the Shanghai Institute of Biochemistry and Cell Biology, Chinese Academy of Sciences. We cultured N2a cells in DMEM (Gibco) supplemented with 10% fetal bovine serum (FBS) (Gibco) and 1% penicillin/streptomycin (Gibco), at 37 °C with 5% $CO_2$ under humidified conditions. Cells were transiently transfected with plasmids and EZ Trans reagent (Shanghai Life-ilab) in a 2:3 ratio according to the manufacturer's protocols 12 h later after being seeded on 12-well plates. For transfection, CasRx plasmid (1500 ng) and gRNA-expressing plasmid (500 ng) were mixed in each well. The control plasmids do not express gRNA.

### Fluorescence-activated cell sorting (FACS)

Approximately 2 days after transient transfection, cells were digested with 0.05% trypsin (Thermo Fisher Scientific, Massachusetts, USA) and sorted by BD FACSAria™ Fusion cytometry (USA) (Fig. S19). GFP and mCherry-double positive cells were kept in DMEM and lysed for qPCR analysis.

### Stable cell line culture

The vector HP180-U6-sgRNA-CMV-Cas9-CMV-EGFP targeting *Aqp1* (or *Adrb2*) and the vector expressing donor fragment were co-transfected into N2a cells using Lipofectamine 3000 (Thermo Fisher Scientific) and Opti-MEM medium (Gibco), according to the manufacturer's instructions. Two days after transfection, puromycin (YEASEN) was added into the medium and the final concentration was kept at 4 µg/ml. In order to select the correct cells for the successful knock-in donor fragment, puromycin was maintained until the monoclonal cell line was grown. Four days after transient transfection, cells were digested with 0.05% trypsin and sorted by BD FACSAria™ Fusion cytometry (USA). 10,000 GFP-positive cells were collected and placed in a 10 cm cell culture dish, adherent culture as single cells, and the monoclonal growth was observed. About 1 week after FACS, monoclonal cells in 10 cm dishes were selected and transferred to 96-well plates for future culture. When the monoclonal cells in the 96-well plate grew to the point where passage was needed, half of the digested cells were lysed, identified by PCR and sequenced the inserted fragments (all primers are provided in Table S2). The monoclonal cell lines with correct identification results were selected and expanded for subsequent experiments. Meanwhile, we took the mixed cloned cells 7 days after transfection for lysis and qPCR analysis to detect the *Aqp1* (or *Adrb2*) gene expression level.

### Quantitative PCR

Total RNA was extracted from cells using TRIZOL (Invitrogen, USA) and reverse-transcribed to cDNA by HiScript Q RT SuperMix for qPCR (Vazyme, Nanjing, China) according to the manufacturer's instruction. The RT product was added to AceQ Universal SYBR qPCR Master Mix (Vazyme, Nanjing, China) and quantitative PCR was performed using Roche 480 II-A. The primers used for qPCR are shown in Table S3.

### Animals

Eight-week-old *C57BL/6J* female mice were purchased from Vital River Laboratories, Beijing, China. Mice were housed under controlled conditions of temperature (21 °C to 26 °C), humidity (40–70%) and maintained on a 12 h light/12 h dark cycle with food and water libitum. All mice were fed until 3 months of age and randomly assigned to the corresponding treatments. All animals were euthanized with $CO_2$. All animal experiments were approved by the Ethics Committee for Animal Studies of Eye & ENT Hospital of Fudan University. All methods were carried out in accordance with the relevant guidelines and regulations and are reported in accordance with ARRIVE guidelines.

## Intraocular pressure measurement

IOP was immediately measured under anesthetic conditions (2.5% isoflurane and 0.8 L/min oxygen) using the TonoLab rebound tonometer (Icare, Vantaa, Finland) according to the manufacturer's instructions. Both day (9:00–12:00) and night-time (20:00–23:00) IOPs were monitored once a week throughout the treatment periods as well as baseline IOP was measured before the treatment. The nighttime IOP was measured in darkness for at least 6 h after dark field treatment. The IOP measurements were completed within 4 min in order to avoid the effect of isoflurane on the IOP. Secure the subject and use enough sedation. Bring the tonometer near to the subject's eye fixing the tonometer with hands and/or to some solid object. The central groove should be in a horizontal position. The distance should be 1–4 mm (1/8 inch) from the tip of the probe to the cornea of the eye. Measure takes place by lightly pressing the measurement button. The tip of the probe should contact the central cornea. Six measurements are made consecutively. Press the measurement button carefully, to avoid shaking the tonometer. An average of six IOP readings were taken at each time period and recorded in a masked manner. If the measurements' standard deviation is clearly greater than normal according to the instructions, new measurement is recommended. 0.5% Levofloxacin eye drops (Santen Pharmaceuticals, Japan) was applied topically immediately following measurement.

## Intravitreal injection

Mice were anesthetized using an intraperitoneal injection of ketamine/xylazine (ketamine 120 mg/kg and xylazine 10 mg/kg) (Aladdin, China). In addition, Proparacaine Hydrochloride eye drops (s.a.Alcon-Couvreur n.v., Belgium) were used before treatment. Pupils were dilated with a single drop of Tropicamide Phenylephrine eye drops (Santen Pharmaceuticals, Japan) prior to injection. All intravitreal injection volumes used were 1.5 µl and were delivered using an operating microscope and a 33 G flat head needle with a Hamilton microsyringe (Hamilton Company, Reno, NV, USA). For intravitreal injection, a 29 G needle was used to make a small incision in the eyeball behind the ora serrata and a 33 G Hamilton micro syringe was inserted through the incision to inject AAV or other solution was injected into the vitreous cavity. Ofloxacin eye ointment (Sinqi Pharmaceutical, Shenyang,China) was applied topically immediately following injection.

## Intracameral injection

Mice were anesthetized using an intraperitoneal injection of ketamine/xylazine (ketamine 120 mg/kg and xylazine 10 mg/kg) (Aladdin, China). In addition, Proparacaine Hydrochloride eye drops (s.a.Alcon-Couvreur n.v., Belgium) were used before treatment. All intracameral injection volumes used were 1.5 µl and were delivered using an operating microscope and a 33 G flat head needle with a Hamilton microsyringe (Hamilton Company, Reno, NV, USA). For intracameral injection, a 29 G needle was used to make a small incision at the corneal margin and a 33 G Hamilton micro syringe was inserted through the incision to inject AAV. Ofloxacin eye ointment (Sinqi Pharmaceutical, Shenyang,China) was applied topically immediately following injection.

## Experimental mouse models of ocular hypertension

To elevate the IOP, 3 month-old mice were anesthetized with a solution of tribromoethanol. Proparacaine Hydrochloride eye drops were used before surgery. The microbead ocular hypertension model followed the published protocol[34]. In brief, after anesthesia and pupil dilation, $2.4 \times 10^6$ sterile magnetic microbeads of 4.5 µm in diameter (Dynabeads® M-450 Epoxy, Thermo Fisher Scientific, UK) were injected into the anterior chamber of one eye using a micromanipulator and microsyringe pump (RWD Life Science, China). Meanwhile, the magnets were used to distribute microbeads evenly around the circumference of the anterior chamber using magnets. Ofloxacin eye ointment was applied topically after the injection.

The glucocorticoid-induced ocular hypertension model was performed as previously described using dexamethasone acetate in a vehicle suspension formulation[42,43]. After measuring the baseline IOP, 10 µl of DEX-Ac (Aladdin, China) suspension or normal saline was injected under the fornix conjunctiva of both eyes with a 29 G insulin needle every 7 days for the duration of each experiment. Ofloxacin eye ointment was applied topically after injection. The IOP was checked before the next injection or establishment of the mouse models of ocular hypertension. AAV treatment was randomly distributed to molding eyes. AAV injections were administered 3 days after the induction of the ocular hypertension model.

## Latanoprost eye drops

The mice were given Latanoprost (0.005%, Pfizer Manufacturing Belgium NV) every night for three days after modeling. Mice were fixed for ten seconds after receiving one drop of Latanoprost per eye.

## Immunofluorescence and microscopy

For EGFP immunofluorescent imaging, eyes with the attached optic nerve segment, surgically removed from mice, were fixed in 4% paraformaldehyde, frozen in optical cutting temperature compound (SAKURA, USA). Samples were sectioned (20 µm thick), stained with DAPI (Abcam, UK) and imaged with Zeiss LSM880 microscope suing a 20 × lens.

For target genes immunofluorescent imaging, eyes were surgically removed from mice and fixed in 4% paraformaldehyde (PFA). Samples were frozen in OCT and divided into sections (20µm thick). Tissue sections were blocked (donkey serum and 0.2% Triton-X-100) for 2 h in a dark and humid chamber, then washed briefly with PBS. Tissue sections were incubated with primary antibodies overnight at 4°C and washed 3 × for 5 min each with PBS before 2 h of incubation with secondary antibodies at room temperature. After that, samples were again washed 3 × for 5 min each with PBS and then mounted with mounting medium containing DAPI nuclear stain (Abcam, UK). Tissue sections incubated without primary antibodies were used as negative controls. Primary antibodies used: rabbit anti-beta 2 adrenergic receptor (1:100; Cat#ab182136; Abcam, UK), rabbit anti-aquaporin 1(1:500; Cat#ab300463; Abcam, UK), ROCK1 (E2G4N) Rabbit mAb (1:50; Cat#28999 S; Cell Signaling Technology, USA), ROCK2 (E5T5P) Rabbit mAb (1:50; Cat#47012 S; Cell Signaling Technology, USA), RCVRN Rabbit pAb (1:200; Cat#A6404;ABclonal, China), anti-Arrestin-C (1:200; Cat#ab15282; Sigma-Aldrich, USA) and mouse anti-Flag (1:4000; Cat#F3165; Sigma-Aldrich, USA). Secondary antibodies used: Alexa Fluor®Cy3 AffiniPure Donkey Anti-Rabbit IgG (1:400; Cat#160185; Jackson ImmunoResearch Labs, USA) and Alexa Fluor®Cy5 AffiniPure Donkey Anti-Mouse IgG (1:400; Cat#145170; Jackson ImmunoResearch Labs, USA). Confocal images were acquired using a Zeiss LSM 880 microscope using a 20× lens. The immunofluorescence staining images of retinal photoreceptors were acquired using Olympus APX100 with a 20x lens. Semi-quantitative fluorescence analysis was performed using the software ImageJ (https://imagej.net/ij/index.html).

For ganglion cell counts, eyes with the attached optic nerve segments surgically removed from mice were fixed in 4% PFA for 1 h. The retinas were dissected out for whole-mount staining. Retinal whole-mounts were blocked in the donkey serum and 0.2% Triton-X 100 overnight at 4 °C. After washing with PBS for 5 min, retinas were incubated with RBPMS rabbit polyclonal antibody (1:100; Cat#15187-1-AP; Proteintech, USA) overnight at 4 °C and washed 3 × for 5 min each with PBS before 2 h of incubation with secondary antibodies at room temperature. Retinas were washed 3 × for 5 min each with PBS again and then mounted with DAPI. Secondary antibodies used: Alexa Fluor®488 AffiniPure Donkey Anti-Rabbit IgG (1:200; Cat# 156556;

Jackson ImmunoResearch Labs, USA) and Alexa Fluor®Cy3 AffiniPure Donkey Anti-Rabbit IgG (1:200). For counting RGCs, approximately 9–12 non-overlapping images from the entire retina were captured at 200 × magnification using a Leica DM4B microscope and RBPMS-positive cells were counted using the software ImageJ (https://imagej.net/ij/index.html).

For safety assessment, the brains, hearts, livers, spleens, lungs and kidneys were removed from the mice and fixed with 4% PFA overnight, and kept in 30% sucrose for at least 12 h. Samples were sectioned after embedding and freezing, and slices with the thickness of 20 μm were used for immunofluorescence staining. The brain sections were rinsed thoroughly with PBS. Primary antibodies used: mouse anti-Flag (1:4000). Secondary antibodies used: Alexa Fluor®Cy5 AffiniPure Donkey Anti-Mouse IgG (1:400). After antibody incubation, the slices were washed and covered with a mounting medium containing a DAPI nuclear stain. Confocal images were acquired using a Zeiss LSM 880 microscope using a 20 × lens.

## Histopathological analysis
The brains, hearts, livers, spleens, lungs and kidneys were removed from the mice and fixed with 4% PFA. The eyes used to observe the structure of the anterior segment were fixed with 4%PFA, and the eyeballs used to observe the retinal were immersed in FAS fixation solution. All samples were sent to the Servicebio (Shang Hai, China) for paraffin section, Hematoxylin-Eosin (H&E) staining and TUNEL test. All the images for histopathological analysis were acquired using Olympus APX100 with a 20 x lens.

## Single-cell RNA data alignment and quantification
The glucocorticoid-induced ocular hypertension model was prepared for single-cell RNA sequence. Ten weeks after mold and virus injection, eyes were surgically removed from the mice and trimmed, then surgical scissors were used to take samples 1 mm before and after the ora serrata immediately and kept in a special preservation solution. All operations were performed on ice. The samples were delivered to CapitalBio Technology (Beijing, China) and sequenced for analysis.

Our research methodology primarily involves the alignment and quantification of single-cell RNA sequence data, followed by a series of downstream analyses, including filtering low-quality cells and data integration, cluster annotation and differential gene expression identification, Hallmark pathway and transcription factor (TF) activity assessment.

Sequencing sequences from each sample were aligned with the reference genome provided by 10 x Genome (San Francisco, CA). The Cellranger v7.1.0 analysis pipeline was utilized with default parameters for initial data filtering and quantification. The filtered data was then used for subsequent downstream analyses.

## Filtering of low-quality cells and data integration
Next, we adopted a more stringent filtering strategy. Specifically, cells from the Cellranger output expressing <300 genes or >2500 genes, as well as those with >7500 RNA mapping to the transcriptome and mitochondrial gene expression exceeding 10%, were excluded. Furthermore, DoubletFinder v2.0.3 was employed to identify and filter doublets (DoubletFinder: Doublet Detection in Single-Cell RNA Sequencing Data Using Artificial Nearest Neighbors). After completing these filtering steps, the canonical correlation analysis (CCA) algorithm from Seurat v4.3.0 was used to batch-correct and integrate the three datasets (Integrated analysis of multimodal single-cell data).

## Cluster annotation and differential gene expression identification
We used Seurat to conduct a Principal Component Analysis (PCA) on the integrated data, identifying variable genes ($n = 2000$) and

performing cluster identification, ultimately recognizing 16 distinct clusters. Subsequently, we employed the t-SNE method for dimensionality reduction and visualization. Marker genes reported in the literature (including Pde6a, Nr2e3, Rpe65, among 26 others) aided us in annotating these 16 clusters. Clusters 0 and 1 which expressed Pde6a and Nr2e3 were defined as "rods"[68,69] (Fig. 5c, Fig. S6a–c). Clusters 2 and 3 which expressed Rpe65, were defined as "RPE"[70] (Fig. 5c, Fig. S6a–c). Clusters 4, which expressed Nkg7 and Cd3d, were defined as "Natural killer (NK) cells"[71] (Fig. 5c, Fig. S6a–c). Clusters 5, which expressed S100a8 and S100a9, were defined as "Neutrophils"[72] (Fig. 5c, Fig. S6a–c). Clusters 6, which expressed Sncg, was defined as "retinal ganglion cells (RGC)"[73] (Fig. 5c, Fig. S6a–c). Clusters 7, which expressed Serpine2, were defined as "uveal fibroblasts (U-Fib) cells"[74] (Fig. 5c, Fig. S6a–c). Clusters 8, which expressed H2-Ab1, were defined as "rM Macrophage cells"[75,76] (Fig. 5c, Fig. S6a–c). Clusters 9, which expressed Ptn, were defined as "Müller glia (MG) cells"[77] (Fig. 5c, Fig. S6a–c). Clusters 10, which expressed Slc4a4 and Mmp3, were defined as "scleral fibroblasts (S-Fib) cells"[74] (Fig. 5c, Fig. S6a–c). Clusters 11, which expressed Arr3 and Opn1sw, were defined as "Cones"[69,78] (Fig. 5c, Fig. S6a–c). Clusters 12, which expressed Cd79a, were defined as "B cells"[79] (Fig. 5c, Fig. S6a–c). Clusters 13, which expressed Cspg4 and Acta2, were defined as "CE cells"[80] (Fig. 5c, Fig. S6a–c). Clusters 14, which expressed Plp1, were defined as "Schwamm cells"[81] (Fig. 5c, Fig. S6a–c). Clusters 15, which expressed Rbp2, were defined as "Melanocyte cells"[82] (Fig. 5c, Fig. S6a–c). Additionally, we used the Findmarker function to identify differentially expressed genes between different groups that exhibit |log2FC | >1 and $P < 0.05$. For these differentially expressed genes, we applied clusterProfiler v4.6.2 for Gene Ontology (GO) and Kyoto Encyclopedia of Genes and Genomes (KEGG) enrichment analysis (clusterProfiler 4.0: A universal enrichment tool for interpreting omics data).

## Hallmark pathway and transcription factor activity assessment
Lastly, we obtained the Hallmark gene set from the MSigDB (https://www.gsea-msigdb.org/gsea/msigdb) database and used AUCell v1.20.2 to assess the activity of the Hallmark gene set in each cell. Meanwhile, to evaluate the differences in transcription factor activity between the control and experimental groups, we employed the pySCENIC v0.12.1 pipeline to assess the expression level of TFs in each cell (a scalable SCENIC workflow for single-cell gene regulatory network analysis).

## Statistical analysis
Statistical analysis of the results performed using GraphPad Prism (GraphPad PRISM, Version 8.0), except for IOP measurement. A two-tailed, unpaired Student's t-test was used for comparison, the confidence interval is 95%, and the difference was considered significant when the $P$-value was <0.05. No adjustments were made for multiple comparisons. IOP monitoring results are shown as the mean ± SEM, and for IOP data consisting of more than two groups, varying in a single factor, one-way ANOVA and Bonferroni comparison test were performed using IBM SPSS Statistics 26 (Chicago, USA), the confidence interval is 95%, and differences were considered significant when $P$-values were <0.05. For RGCs counting, data were excluded if the sample quality is poor such as sample defect. No data were excluded from other analyses.

## Reporting summary
Further information on research design is available in the Nature Portfolio Reporting Summary linked to this article.

# Data availability
The single-cell RNA sequencing data GSE24776 generated in this study have been deposited in the NCBI GEO database Source data are provided with this paper.

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

## Acknowledgements

We would like to thank Qiqi Zhang for schematic design and visualization of this manuscript. This work was supported in part by the Project of National Natural Science Foundation of China (Grant No. 82271048 and 80301176); Shanghai Science and Technology (Grant No. 22S11900200, 23XD1420500); EYE & ENT Hospital of Fudan University High-level Talents Program (Grant No. 2021318); Clinical Research Plan of SHDC (Grant No. SHDC2020CR1043B); Project of Shanghai Xuhui District Science and Technology (Grant No. 2020-015); Program for Professor of Special Appointment (Eastern Scholar, TP2022046) at Shanghai Institutions of Higher Learning; Medical Engineering fund of Fudan University (yg2023-06, yg2023-26).

## Author contributions

X.Z. and J.H. jointly conceived the project. M.Y. and Z.Z. jointly designed experiments. M.Y., Z.C., X.C., Q.G. performed in vitro knockdown efficiency verification and plasmid construction. M.Y., Z.Z., G.Z., A.C., Z.L., Y.W., R.N. conducted most of animal experiments, including screening for the AAV serotypes, establishment of mouse model, and verifying the effect of reducing IOP in model mice. M.Y., Z.C., X.C., Q.G., and S.L.

conducted a series of experiments to assess safety. Z.Z. and Z.Z. performed the droplet-based scRNA-seq. Z.C., X.C., Q.G., G.Z., A.C., Z.L., Y.W., R.N., and C.M., assisted with experiments. X.Z. and J.H. supervised the whole project. M.Y. and Z.Z. wrote the manuscript with data contributed by all author who participated in the project.

## Competing interests

Outside the submitted work, C.M. has received consultancy fees/honorarium/travel support (past 36 months) from: Acufocus (Irvine, California, USA), Atia Vision (Campbell, California, USA), Bausch and Lomb (Bridgewater, New Jersey, USA), Bayer (Leverkusen, Germany), British Society of Refractive Surgery (Oxford, UK), BVI (Liège, Belgium), Coopervision (Pleasanton, California, USA), Cutting Edge (Labége, France), Hoya (Frankfurt, Germany), Knowledge Gate Group (Copenhagen, Denmark), Johnson & Johnson Surgical Vision (Santa Ana, California, USA), Keio University (Tokyo, Japan), Medevise Consulting SAS (Strasbourg, France), Ophtec BV (Groningen, The Netherlands), Portuguese Society of Ophthalmology (Coimbra, Portugal), ROHTO (Tokyo, Japan), Royal College of Ophthalmologists (London, UK), SightGlass vision (Menlo Park, California, USA), Science in Vision (Bend, Oregan, USA), Scope (Crawley, UK), SpyGlass (Aliso Viejo, California, USA), Sun Yat-sen University (Guangzhou, China), Thea pharmaceuticals (Clemont-Ferrand, France), Vold Vision (Arkansas, USA). C.M. developed the Quality of Vision (QoV) questionnaire and the Orthokeratology and Contact Lens Quality of Life Questionnaire (OCL-QoL), and has a financial interest in these tools. He also consults on topics including Rasch analysis, questionnaires, statistical analyses, and clinical/surgical ophthalmology topics. C.M. is a co-applicant on an awarded Welsh Government research grant related to diabetic eye disease (unpaid role), council member of the British Society for Refractive Surgery (unpaid role) and a PROM advisor to the Royal College of Ophthalmologists (unpaid role). C.M. has undertaken paid peer reviews for Research Square (Durham, North Carolina, USA). All other authors declare no competing interests.
