## [Peer Review File · Nature Communications]

REVIEWER COMMENTS

Reviewer #1 (Remarks to the Author):

This study by Yao and colleagues use a novel CRISPR based approaches to disrupt the expression of Rock 1 and Rock 2 as well as Aqp1 and Adrb2 to enhance outflow in the anterior segment of mouse models of glaucoma and therefor decrease IOP. It is a highly novel approach that seems to produce robust decreases in IOP in 2 verified models of a disease. While the study is well planned a out and has novel molecular biology, as a translational study, it lacks some critical data. Outlined below, are comments that are focused largely on these deficits and are essential for a study of this sort. There is very little data that can convince the reader on the overt retinal and anterior segment safety of this approach and these would need to be addressed.

Major points

Although the authors state that they used both IVT and IC injection of the AAV, it would be important to show a comparison of both approached as well as the safety profile using histopathological images. Maybe I missed it, but I can only see data from IVT injections.

It would be important to show histopathology images of the anterior segment and retina in mice injected with the targeting vectors. This is critical in allowing for the reader to appreciate the perceived safety of the approach.

Supplemental Figures 4 and 5 need H&E stained sections as there is very little observable detail in DAPI stained images.

AAV shH10 serotype robustly transduced the retina (see here: PMID: 19826483). What effect does casRx expression have in the retina. Have authors conducted ERG analysis and histopathological assessment of the retina at various time points post IVT injection? Wouldn't anterior segment injection be more amenable to this therapy?

How does this therapy compare to Latanoprost? This should be an additional positive control group.

Discussion

The discussion is very poorly referenced.

The concept of compensatory AH recovery when focusing on a single gene needs to be cited.

The whole section on neutrophils in glaucoma is without any reference.

Reviewer #2 (Remarks to the Author):

The authors presented an intriguing approach utilizing the CRISPR CasRx gene editing system to knock down Rock1 and Rock2, as well as Aqp1 and Adrb2, aiming to develop a novel treatment for elevated intraocular pressure (IOP). They created two distinct mouse models: the magnetic microbead occlusion model and the glucocorticoid-induced model. Using these, they demonstrated significant reductions in intraocular pressure through vitreous injection of these molecules. The findings are both promising and impressive. However, one area of concern is the measurement of intraocular pressure (IOP) in mice, which is challenging and requires skilled technicians. The authors should add a further description for IOP measurement, specifically focusing on a very minimal standard deviation.

Furthermore, the authors evaluated the effectiveness of LacZ virus versus shH10 Y25/26 virus, finding the latter more effective. Their strategy of targeting four genes for gene therapy is unique and promising.

There is, however, some confusion regarding the subtitle and related content starting from line 238. The authors suggest that blocking ROCK inhibitors increases neutrophil infiltration, contributing to reduced IOP through immune-mediated reactions. This interpretation contrasts with a general understanding of the beneficial effects of ROCK inhibitors, possibly indicating species-specific differences between mice and humans.

Lastly, given the important role of Aqp1 in aqueous humor secretion from the ciliary body and water flow from corneal endothelial cells to the anterior chamber, assessing any adverse effects on

corneal endothelial cells when injecting compounds into the anterior chamber is crucial. The authors may add some comments regarding this subject in the manuscript.

Reviewer #3 (Remarks to the Author):

In this manuscript, Yao et al. propose gene therapy for glaucoma treatment by targeting multiple genes using a CRISPR-CasX viral approach. They simultaneously target the expression of Rock1 and 2 genes, associated with the aqueous humor outflow through the trabecular mesh (TM) and the genes Aqp1 and Adrb2 related to the generation of aqueous humor by the ciliary body. By doing this they observed reduced intraocular pressure (IOP) and protection of retinal ganglion cells, thereby claiming delayed disease progression.

This manuscript provides additional evidence to previous studies from different labs in which reducing expression of these genes or pharmacological inhibition of Rock activity results in decreased IOP and preserves RGC integrity.

Indeed, there is a vast body of literature supporting the use of Rock inhibitors (RKI) (some are already in clinics e.g. Netarsudil, Ripasudil, etc) to manage glaucoma. By preventing actomyosin contraction RKIs cause TM cells to relax leading to the increase of intercellular space, disrupt focal adhesions in the TM and the inner wall endothelial lining of the Schlemm canal and, as a consequence, facilitate AH outflow and decrease IOP. However, regular use of RKI in drops have undesirable side effects and therefore the use of gene therapy to help controlling IOP with one single eye injection would be an interesting therapeutic option.

Previous work also suggests that silencing the beta 2 adrenergic receptor (Adrb2) with siRNA (SYL040012) is effective in reducing aqueous humor production at the CB, and as a result IOP, and a recent report (Wu et al., Translation Medicine, 2020), which the authors mention, show that ciliary body aquaporin 1 disruption using Crispr-Cas9 results in reduced IOP.

The findings reported in this work are not novel and based on the available literature are somehow expected. Saying that the manuscript is well structured, technically sound, reads well and the findings are presented with clarity and the experimental data mostly sustains the conclusions made by the authors.

I have some queries:

1. Based on the experimental protocol, where viral injections are carried out 3 days after the induction of IOP, the authors can only claim that their gene therapy approach can prevent the hallmarks of the disease and protect RGC from increasing IOP. To claim that their gene therapy approach can delay disease progression would require to inject the eye at a time where the

hallmarks of the disease IOP, RGC death, etc are already well established, i.e., weeks after the induction of the glaucoma model. These experiments would strengthen their claims and should be at least discussed by the authors.

2.The authors chose to use a glucocorticoid-induced glaucoma model (by injecting dexamethasone into the eye every 7 days for the duration of the experiment) to carry out single cell transcriptomic analyses. Their sc results show that most of DE genes up-regulated from Y25 relative to controls were mainly related to immune processes. Knowing that dexamethasone and glucocorticoids in general are powerful inhibitors of immune cell function and can modulate the numbers of NK cells, among other immune cells, are they not afraid that this model will mask inflammation processes that otherwise would be much higher “in real life”? In terms of the sc analyses how does this model would compare with the microbead ocular hypertension model? And is there a rationale to why the authors did not use this model instead of the GC one?

Minor:

3. In addition to RGC data do the authors have any data concerning neuronal loss in the the optic nerve? Or OCT data?

RESPONSE TO REVIEWER COMMENTS

Reviewer #1 (Remarks to the Author):

This study by Yao and colleagues use a novel CRISPR based approaches to disrupt the expression of Rock 1 and Rock 2 as well as Aqp1 and Adrb2 to enhance outflow in the anterior segment of mouse models of glaucoma and therefor decrease IOP. It is a highly novel approach that seems to produce robust decreases in IOP in 2 verified models of a disease. While the study is well planned a out and has novel molecular biology, as a translational study, it lacks some critical data. Outlined below, are comments that are focused largely on these deficits and are essential for a study of this sort. There is very little data that can convince the reader on the overt retinal and anterior segment safety of this approach and these would need to be addressed.

Reply: Thank you very much for your thoughtful comment and valuable suggestions. We are truly grateful for your interest in our work. This work demonstrates a new approach using the CRISPR CasRx RNA editing system delivered by AAV to knock down *Rock1* and *Rock2*, as well as *Aqp1* and *Adrb2*, aiming to reduce intraocular pressure (IOP), which may pay a way for a new type of glaucoma treatment. Following your suggestion, a range of experimental work was performed. The subsequent findings have been meticulously included in the revised manuscript and accompanying supporting information, which have been highlighted in red font for your convenience. Once again, we sincerely appreciate your interest and feedback on our work. For ease of reading, your comments have been categorized into seven sections and addressed individually. The detailed responses are presented as follows:

1. Although the authors state that they used both IVT and IC injection of the AAV, it would be important to show a comparison of both approached as well as the safety profile using histopathological images. Maybe I missed it, but I can only see data from IVT injections.

Reply: Thank you for your valuable comment. In fact, the study described in this paper only discussed the infection of various parts of the eye through IVT injection of different types of AAV, and the results are illustrated in Fig. 2. The relevant content of

the discussion section has been corrected and marked in red font. However, IVT and IC are both important and commonly used ways to inject AAV, it is indeed necessary to compare the two modes of injection. Compared with IC injection of shH10 virus, previous experiments conducted by our team have shown that the shH10 virus can stably infect the ciliary body and trabecular meshwork of the eye for a long time through IVT injection. The corresponding confocal images are presented in Fig. S13. For the intravitreal injection group, the strong GFP signal can be detected at all three time points, while in the intracameral injection group, significant GFP signal was observed only in the samples taken at the first week. This may be related to IC injection causing more virus loss along the outflow channel of aqueous humor. Based on the results of this previous study, in order to obtain a more stable effect of AAV infection, we chose the IVT method for virus injection in this study.

Following your recommendation, we used histopathological analysis to evaluate the safety of IVT injection, including hematoxylin and eosin (H&E) staining of the eyeballs ten weeks after AAV injection and TUNEL staining to detect apoptotic cells. The results of our evaluation were presented in the revised manuscript and accompanying supporting information (Fig. S14) as below, with key findings highlighted in red font. The results of H&E staining of the cornea, anterior chamber angle and retina as well as TUNEL staining showed no obvious differences between the wild type mice and the mice intravitreal injected with shH10 Y25/Y26 virus. H&E staining images (Fig. S14a) show that ten weeks after IVT injection of shH10 virus, all cells evenly distributed in both cornea, iris, TM, ciliary body, retina, and no obvious cell damage or tissue injury was observed. Images of TUNEL test are shown in Fig. S14b, no green color can be observed in cornea, iris, TM, ciliary body or retina of the eyes IVT injected with AAV as well as wild type group, indicating that both shH10 Y25 and shH10 Y26 virus did not induce apoptosis. These histopathological results confirming that it is safe and feasible to inject targeted shH10 virus through intravitreal injection.

Fig. S13. Intraocular infection of shH10 virus by different injection methods.

The injected virus was shH10-EFS-EGFP with titer of $1E+13$ GC/ml and injection dose of $1.5\mu\text{l}$. The eyeball was taken at one week, two weeks and three weeks after injection, respectively, and the expression of the virus in the ciliary body and trabecular reticulum was observed. The arrow indicated the ciliary body and the box indicated the trabecular meshwork.

a. Confocal images of ocular infection in mice after intravitreal injection of shH10 AAV virus. Scale bars, $100\mu\text{m}$.

b. Confocal images of ocular infection in mice after intracameral injection of shH10 AAV virus. Scale bars, $100\mu\text{m}$.

Fig. S14. Ocular safety of intravitreal injection of virus.

a. Hematoxylin-Eosin staining (H&E) of the cornea, anterior chamber angle and retina

in mice intravitreal injected with shH10 Y25/Y26 virus for 10 weeks. WT: wild type mice. Scale bars,100 μ m.

b. TUNEL test of cornea, anterior chamber angle and retina after being intravitreal injected with shH10 Y25/Y26 virus for ten weeks. WT: wild type mice. Scale bars,100 μ m.

2. It would be important to show histopathology images of the anterior segment and retina in mice injected with the targeting vectors. This is critical in allowing for the reader to appreciate the perceived safety of the approach.

Reply: Thanks for your instrumental suggestion. Following your comment, we used histopathological analysis to evaluate the safety of IVT injection, including hematoxylin and eosin (H&E) staining of the eyeballs ten weeks after AAV injection and TUNEL staining to detect apoptotic cells. The results of our evaluation were presented in the revised manuscript and accompanying supporting information (Fig. S14) as below, with key findings highlighted in red font. The results of H&E staining of the cornea, anterior chamber angle, and retina as well as TUNEL staining showed no obvious differences between the wild type mice and the mice intravitreal injected with shH10 Y25/Y26 virus. H&E staining images (Fig. S14a) show that ten weeks after IVT injection of shH10 virus, all cells evenly distributed in both cornea, iris, TM, ciliary body, retina, and no obvious cell damage or tissue injury was observed. Images of TUNEL test are shown in Fig. S14b, no green color can be observed in cornea, iris, TM, ciliary body or retina of the eyes IVT injected with AAV as well as wild type group, indicating that both shH10 Y25 and shH10 Y26 virus did not induce apoptosis. These histopathological results confirming that it is safe and feasible to inject targeted shH10 virus through intravitreal injection.

Fig. S14. Ocular safety of intravitreal injection of virus.

a. Hematoxylin-Eosin staining (H&E) of the cornea, anterior chamber angle and retina

in mice intravitreal injected with shH10 Y25/Y26 virus for 10 weeks. WT: wild type mice. Scale bars,100 μ m.

b. TUNEL test of cornea, anterior chamber angle and retina after being intravitreal injected with shH10 Y25/Y26 virus for ten weeks. WT: wild type mice. Scale bars,100 μ m.

3. Supplemental Figures 4 and 5 need H&E stained sections as there is very little observable detail in DAPI stained images.

Reply: Thanks for your kindly comment. According to your suggestion, H&E staining results of multiple organs in each group were added on the basis of previous experiments. At ten weeks post AAV injection, multiple tissues, including the brain, heart, liver, spleen, lung, and kidney were collected, with wild type mice and model mice injected with saline serving as control. As depicted in Fig. S15, the results demonstrated that there were no observable pathological changes in the brain, heart, liver, spleen, lung, or kidney over the course of treatment. Specifically, hepatocytes appeared normal in the liver samples, no pulmonary fibrosis was detected in the lung samples, the glomerulus structure was clearly visible in the kidney sections, and there was no evidence of necrosis in any of the histological samples. These findings strongly suggest that IVT injection of shH10 Y25 virus and shH10 Y26 virus have minimal side effects and possess great clinical applicability as a gene therapy to reduce intraocular pressure.

Fig. S15. Multi-organ safety of intravitreal injection of virus.

a. Hematoxylin-Eosin staining (H&E) of the different organs of the magnetic beads-induced model mice intravitreal injected with shH10 LacZ/Y25/Y26 virus. All samples were taken 10 weeks after modeling. WT: wild type mice. Scale bars,100 μ m.

b. Hematoxylin-Eosin staining (H&E) of the different organs of the DEX-induced model mice intravitreal injected with shH10 LacZ/Y25/Y26 virus. All samples were taken 10 weeks after modeling. WT: wild type mice. Scale bars,100 μ m.

4. AAV shH10 serotype robustly transduced the retina (see here: PMID: 19826483). What effect does casRx expression have in the retina. Have authors conducted ERG analysis and histopathological assessment of the retina at various time points post IVT injection? Wouldn't anterior segment injection be more amenable to this therapy?

Reply: Thank you for your valuable comment. According to the article (PMID: 19826483) and our previous experiment, AAV shH10 serotype can indeed transduce the retina, and the CasRx system delivered by the AAV may have some effects on the retina's target gene. Therefore, how to accurately deliver gene editing systems to target tissues and target cells remain a huge challenge in the field of gene therapy. In subsequent studies, we will improve the delivery regimen by screening for tissue-specific promoters or evolving AAV serotypes to avoid effects on non-target tissues. In this study, our team performed ERG analysis on mice at one week, two weeks, and three weeks after IVT injection of AAV to determine the effects of shH10 Y25/26 virus on visual function. Waveforms were evaluated for negative a-wave (photoreceptor responses) and positive b-wave (cone and rod system responses) amplitudes (Fig. S16a). As shown in Fig. S16b-c, the amplitudes of a-waves and b-waves were no significant difference in AAV treated eyes compared to wild type control eyes (WT). Furthermore, Immunofluorescence staining was performed on retinal photoreceptors of mice post ten weeks of shH10 Y25/Y26 virus IVT injection. There was no significant difference in cone and rod cells of the AAV treated eyes compared to the WT group. Meanwhile, we used histopathological analysis to evaluate the safety of IVT injection, including hematoxylin and eosin (H&E) staining of the retina ten weeks after AAV injection and TUNEL staining to detect apoptotic cells. The results of H&E staining of the retina as well as TUNEL staining showed no obvious differences between the wild type mice and the mice intravitreal injected with shH10 Y25/Y26 virus (Fig. S14). H&E staining images (Fig. S14a) show that all cells evenly distributed in retina and no obvious cell damage or tissue injury was observed. Images of TUNEL test are shown in Fig. S14b, no green color can be observed in retina of the eyes IVT injected with AAV as well as wild type group, indicating that both shH10 Y25 and shH10 Y26 virus did not induce apoptosis. These results confirm that IVT injection of shH10 Y25/26 virus have no

serious effect on vision function.

Compared with intracameral injection of shH10 virus, previous experiments conducted by our team have shown that the shH10 virus can stably infect the ciliary body and trabecular meshwork of the eye for a long time through IVT injection. The corresponding confocal image are presented in Fig. S13. For the intravitreal injection group, the strong GFP signal can be detected at all three time points, while in the intracameral injection group, significant GFP signal was observed only in the samples taken at the first week. This may be related to IC injection causing more virus loss along the outflow channel of aqueous humor. Based on the results of this previous study, in order to obtain a more stable effect of AAV infection, we chose the IVT method instead of anterior segment injection for virus injection in this study.

Fig. S13. Intraocular infection of shH10 virus by different injection methods.

The injected virus was shH10-EFS-EGFP with titer of $1E+13$ GC/ml and injection dose of $1.5\mu\text{l}$. The eyeball was taken at one week, two weeks and three weeks after injection, respectively, and the expression of the virus in the ciliary body and trabecular reticulum was observed. The arrow indicated the ciliary body and the box indicated the trabecular meshwork.

a. Confocal image of ocular infection in mice after intravitreal injection of shH10 AAV

virus. Scale bars,100 μ m.

b. Confocal image of ocular infection in mice after intracameral injection of shH10

AAV virus. Scale bars,100 μ m.

Fig. S14. Ocular safety of intravitreal injection of virus.

a. Hematoxylin-Eosin staining (H&E) of the cornea, anterior chamber angle and retina

in mice intravitreal injected with shH10 Y25/Y26 virus for 10 weeks. WT: wild type mice. Scale bars,100 μ m.

b. TUNEL test of cornea, anterior chamber angle and retina after being intravitreal injected with shH10 Y25/Y26 virus for ten weeks. WT: wild type mice. Scale bars,100 μ m.

Fig. S16. Intravitreal injection of shH10 virus did not affect retinal photoreceptor function.

a. Full-field electroretinography (ERG) result of mice intravitreal injected with shH10 Y25/Y26 virus for 10 weeks. WT: wild type mice.

b-c. Quantitative analysis of a-wave (b) and b-wave (c) amplitude. n=6. All values were expressed as mean \pm SEM, unpaired T-test, ns, not significant.

d. Immunofluorescence staining image of retinal photoreceptors of mice intravitreal injected with shH10 Y25/Y26 virus for 10 weeks. Arrestin labels cones. RCVRN labeled rod cells. WT: wild type mice. Scale bars, 50 μ m.

5. How does this therapy compare to Latanoprost? This should be an additional positive control group.

Reply: Thank you very much for your valuable suggestions. Follow your comment, we compared the intraocular pressure reduction effects of shH10 Y25 (EFS-CasRx/U6-*Rock1-Rock2*) virus, shH10 Y26 (AAV-EFs-CasRx/U6-*Aqp1-Adrb2*) virus and Latanoprost on different models of mice within six weeks. The results of our evaluation were presented in the revised manuscript and accompanying supporting information (Fig. S17) as below. In mice with magnetic bead-induced IOP, both two treatment virus achieved better IOP reduction over the long term than Latanoprost. Since the magnetic beads-induced model is more like angle-closure glaucoma, this result is consistent with the indication of Latanoprost. In the mouse model of intraocular hypertension induced by DEX, the effect of treatment virus on reducing IOP was similar to that of Latanoprost. The data from this study provide meaningful information for potential clinical applications.

Fig. S17. The effect of AAV-delivered CasRx system on reducing IOP was compared with Latanoprost

a. Intraocular pressure (IOP) changed during day and night in each group after monocular magnetic beads modeling and injection of shH10 Y25, shH10 Y26 or using Latanoprost drops daily during six weeks. WT group n=8, Beads+Y25 group n=6, Beads+Y26 group n=6, Beads+Latanoprost group n=6, all values were expressed as mean±SEM, and differences between groups were tested by one-way ANOVA.

b. IOP changed during day and night in each group after DEX modeling and injection of shH10 Y25, shH10 Y26 or using Latanoprost drops daily during six weeks. WT group n=8, DEX+Y25 group n=8, DEX+Y26 group n=6, DEX+Latanoprost group n=10, all values were expressed as mean±SEM, and differences between groups were tested by one-way ANOVA.

6. Discussion. The discussion is very poorly referenced. The concept of compensatory AH recovery when focusing on a single gene needs to be cited.

Reply: Thanks for your reminding. Follow your suggestion, we discussed concept of compensatory AH recovery and more references are cited in the revised manuscript

with highlighted in red font.

7. The whole section on neutrophils in glaucoma is without any reference.

Reply: We thank the reviewer for this valuable suggestion. Accordingly, we have added references to support the results of our scRNA-seq analysis. We have also added text to this section to elaborate on our findings in more detail. Several studies have been reported challenging the notion that immune response activation in glaucoma is harmful. In particular, the latest batch of studies found that recruited inflammatory cells in glaucoma may have a role in cleaning up cell debris, cleaning up dead cells and harmful protein products, and thus protecting neurons in neurodegenerative diseases, which provides strong support for our findings.

Reviewer #2 (Remarks to the Author):

The authors presented an intriguing approach utilizing the CRISPR CasRx gene editing system to knock down *Rock1* and *Rock2*, as well as *Aqp1* and *Adrb2*, aiming to develop a novel treatment for elevated intraocular pressure (IOP). They created two distinct mouse models: the magnetic microbead occlusion model and the glucocorticoid-induced model. Using these, they demonstrated significant reductions in intraocular pressure through vitreous injection of these molecules. The findings are both promising and impressive.

Reply: Thank you for your careful review work and valuable comments. We are truly grateful for your interest in our work. This work demonstrates a new approach using the CRISPR CasRx RNA editing system delivered by AAV to knock down *Rock1* and *Rock2*, as well as *Aqp1* and *Adrb2*, aiming to reduce intraocular pressure (IOP), which may pay a way for a new type of glaucoma treatment. For convenience, your comments have been categorized into three sections and addressed individually. The point-by-point response is shown below.

1. However, one area of concern is the measurement of intraocular pressure (IOP) in

mice, which is challenging and requires skilled technicians. The authors should add a further description for IOP measurement, specifically focusing on a very minimal standard deviation.

Reply: Thank you for your valuable comment. According to your recommendation, we add more details of IOP measurement in the revised manuscript. We further explained that“Secure the subject and use enough sedation. Bring the tonometer near to the subject’s eye fixing the tonometer with hands and/or to some solid object. The central groove should be in a horizontal position. The distance should be 1-4 mm (1/8 inch) from the tip of the probe to the cornea of the eye. Measure takes place by lightly pressing the measurement button. The tip of the probe should contact the central cornea. Six measurements are made consecutively. Press the measurement button carefully, to avoid shaking the tonometer. An average of six IOP readings were taken at each time period and recorded in a masked manner. If the measurements’ standard deviation is clearly greater than normal according to the instructions, new measurement is recommended.”

2. Furthermore, the authors evaluated the effectiveness of LacZ virus versus shH10 Y25/26 virus, finding the latter more effective. Their strategy of targeting four genes for gene therapy is unique and promising.

There is, however, some confusion regarding the subtitle and related content starting from line 238. The authors suggest that blocking ROCK inhibitors increases neutrophil infiltration, contributing to reduced IOP through immune-mediated reactions. This interpretation contrasts with a general understanding of the beneficial effects of ROCK inhibitors, possibly indicating species-specific differences between mice and humans.

Reply: We express our gratitude for this invaluable suggestion. Over the past few years, there has been increasing clinical and experimental research evidence indicating the significance of immune response in the pathogenesis of glaucoma. Several studies suggest that the immune response activated in glaucoma patients may be detrimental to neurons. However, it is important to consider that in the immune response to glaucoma, cell survival or death is determined by a balance between positive and negative

regulators¹. Neutrophils and T cells play a role in eliminating dysregulated synapses of retinal ganglion cells (RGCs), phagocytosing dead cells and debris, as well as presenting antigens. Inflammatory states are not only associated with neurotoxic consequences but also have neuroprotective effects. For instance, T-cell-mediated immune responses can initially limit neurodegeneration², while recruitment of T cells facilitates early communication between the immune system and harmful stimuli-induced cell debris.

In rodents, T-cell-mediated autoimmunity exerts protective effects on the progression of glaucomatous optic neuropathy through active or passive immune autoantigens, phagocytosis of deceased neurons, and clearance of toxic aggregates and cellular debris that impact the central nervous system. This mechanism prevents disruption of immune regulation caused by persistent harmful stimuli, thereby reducing secondary degeneration of retinal ganglion cells (RGCs), which is referred to as "protective immunity"³⁻⁶.

We observed an increase in N/K T cells and neutrophils following Y25 virus intervention compared to the control group. Additionally, single-cell sequencing analysis revealed significant up-regulation and down-regulation of numerous genes upon interference with *Rock1* and *Rock2* expression (Fig. S8a). These differentially expressed genes were found to be enriched in immune response-related signaling pathways such as "antigen processing and presentation of endogenous antigen", "cell killing", and "antigen processing and presentation of endogenous peptide antigen via MHC class" (Fig. S8b). Further analysis demonstrated that these pathways are involved in inflammatory responses, including key contributions from neutrophils and N/K T-cell (Fig. S7g-j). Furthermore, recent studies have challenged the previous notion that neutrophil recruitment is detrimental to neurodegeneration, revealing subpopulations of neutrophils with protective and reparative effects on nerves⁷.

However, as raised by the reviewer, establishing a definitive causal relationship to demonstrate that knocking down *Rock1* and *Rock2* can up-regulate protective cytokines in complex inflammatory responses to achieve a new balance is challenging. Building

upon the current findings, it is suggested that the mechanism of reducing IOP through inhibiting *Rock1* and *Rock2* may be regulated via the aforementioned signaling pathways. In conclusion, glaucoma is characterized by chronic changes, and single-cell data currently available only provide a snapshot of its status at one time point. Follow-up studies are necessary to collect samples at multiple time points during treatment to further elucidate the role of immune regulation in glaucoma therapy. The limitations mentioned above could explain the apparent contradictions among different studies while also serving as potential breakthroughs for discovering new theories such as immune regulation as a neuroprotective strategy for treating glaucoma⁶.

3. Lastly, given the important role of *Aqp1* in aqueous humor secretion from the ciliary body and water flow from corneal endothelial cells to the anterior chamber, assessing any adverse effects on corneal endothelial cells when injecting compounds into the anterior chamber is crucial. The authors may add some comments regarding this subject in the manuscript.

Reply: Thanks for your instrumental suggestion. Following your comment, we used histopathological analysis to evaluate the safety of intravitreal injecting shH10 Y26 (AAV-EFs-CasRx/U6-*Aqp1-Adrb2*) virus, including hematoxylin and eosin (H&E) staining of the eyeballs ten weeks after AAV injection and TUNEL staining to detect apoptotic cells. The results of our evaluation were presented in the revised manuscript and accompanying supporting information (Fig. S14) as below, with key findings highlighted in red font. The results of H&E staining of the cornea as well as TUNEL staining showed no obvious differences between the wild type mice and the mice intravitreal injected with shH10 Y26 virus. H&E staining images (Fig. S14a) show that ten weeks after IVT injection of shH10 virus, all cells evenly distributed in cornea and no obvious corneal endothelial cells damage was observed. Images of TUNEL test are shown in Fig. S14b, no green color can be observed in cornea of the eyes IVT injected with shH10 Y26 virus as well as wild type group, indicating that knocking down *Aqp1* did not induce apoptosis. These histopathological results confirming that it is safe and feasible to inject targeted shH10 virus through intravitreal injection.

Fig. S14. Ocular safety of intravitreal injection of virus.

a. Hematoxylin-Eosin staining (H&E) of the cornea, anterior chamber angle and retina

in mice intravitreal injected with shH10 Y25/Y26 virus for 10 weeks. WT: wild type mice. Scale bars,100µm.

b. TUNEL test of cornea, anterior chamber angle and retina after being intravitreal injected with shH10 Y25/Y26 virus for ten weeks. WT: wild type mice. Scale bars,100µm.

Reviewer #3 (Remarks to the Author):

In this manuscript, Yao et al. propose gene therapy for glaucoma treatment by targeting multiple genes using a CRISPR-CasX viral approach. They simultaneously target the expression of Rock1 and 2 genes, associated with the aqueous humor outflow through the trabecular mesh (TM) and the genes Aqp1 and Adrb2 related to the generation of aqueous humor by the ciliary body. By doing this they observed reduced intraocular pressure (IOP) and protection of retinal ganglion cells, thereby claiming delayed disease progression.

This manuscript provides additional evidence to previous studies from different labs in which reducing expression of these genes or pharmacological inhibition of Rock activity results in decreased IOP and preserves RGC integrity.

Indeed, there is a vast body of literature supporting the use of Rock inhibitors (RKI) (some are already in clinics e.g. Netarsudil, Ripasudil, etc) to manage glaucoma. By preventing actomyosin contraction RKIs cause TM cells to relax leading to the increase of intercellular space, disrupt focal adhesions in the TM and the inner wall endothelial lining of the Schlemm canal and, as a consequence, facilitate AH outflow and decrease IOP. However, regular use of RKI in drops have undesirable side effects and therefore the use of gene therapy to help controlling IOP with one single eye injection would be an interesting therapeutic option.

Previous work also suggests that silencing the beta 2 adrenergic receptor (Adrb2) with siRNA (SYL040012) is effective in reducing aqueous humor production at the CB, and as a result IOP, and a recent report (Wu et al., Translation Medicine, 2020), which the authors mention, show that ciliary body aquaporin 1 disruption using Crispr-Cas9 results in reduced IOP.

The findings reported in this work are not novel and based on the available literature are somehow expected. Saying that the manuscript is well structured, technically sound, reads well and the findings are presented with clarity and the experimental data mostly sustains the conclusions made by the authors.

Reply:

Thank you for your professional review work on our manuscript. We are truly grateful for your interest in our work. This work demonstrates a new approach using the CRISPR CasRx RNA editing system delivered by AAV to knock down *Rock1* and *Rock2*, as well as *Aqp1* and *Adrb2*, aiming to reduce intraocular pressure (IOP), which may pay a way for a new type of glaucoma treatment. Once again, we sincerely appreciate your interest and feedback on our work. The point-by-point response is shown below.

I have some queries:

1. Based on the experimental protocol, where viral injections are carried out 3 days after the induction of IOP, the authors can only claim that their gene therapy approach can prevent the hallmarks of the disease and protect RGC from increasing IOP. To claim that their gene therapy approach can delay disease progression would require to inject the eye at a time where the hallmarks of the disease IOP, RGC death, etc are already well established, i.e., weeks after the induction of the glaucoma model. These experiments would strengthen their claims and should be at least discussed by the authors.

Reply: Thank you for your kind comment. According to previous research⁸ (PMID: 18414476), AAV begins to express in vivo 1-2 weeks after injection and reaches its peak expression at 3-4 weeks. We also studied two mouse models of ocular hypertension that mimic glaucoma and examined the changes of IOP. As shown in the Fig. 2e and Fig. 2h, the IOP of both models increased steadily one week after modeling, which was consistent with the time rule of virus expression. However, one of the hallmarks of the disease, RGC death, has not been established. Therefore, according to

your suggestion, we have discussed this in the revised manuscript. In subsequent experiments, we can further explore the effect of AAV injection at different stages of the disease on the prognosis of glaucoma.

Fig. 2. Evaluation of AAV virus infection and establishment of two intraocular hypertension models in mice.

a. Intraocular infection of C57BL/ 6J female mice after intravitreal injection of different serotypes of AAV virus. The injected virus was AAV2-EFS-EGFP or shH10-EFS-EGFP

with titer of $1E+13$ GC/ml and injection dose of $1.5\mu\text{l}$. The eyeball was taken at one week, two weeks and three weeks after injection, respectively, and the expression of the virus in the ciliary body and trabecular reticulum was observed. The arrow indicated the ciliary body and the box indicated the trabecular reticulum, $n=3$. Scale bars, $100\mu\text{m}$.

b. two weeks after AAV2 virus or shH10 virus injected into vitreous cavity, the percentage of EGFP⁺ cells in the trabecular meshwork and ciliary body area in total cells. Data were expressed as mean \pm SEM, * $p < 0.05$, ** $p < 0.01$, *** $p < 0.001$, unpaired T-test.

c. The frozen section of the mouse eyeballs six months after injection of $1.5\mu\text{l}$ shH10-EFS-EGFP virus into the glass cavity showed that the virus was still expressed in the ciliary body and trabecular reticulum, $n=2$, Scale bars, $100\mu\text{m}$.

d. The establishment and evaluation process of two intraocular hypertension mouse models. Weekly injection of dexamethasone acetate suspension through the fornix conjunctiva of both eyes was one of modeling methods. Another is magnetic microbeads induced intraocular hypertension. RGCs of Rbpms⁺ were counted from 6-12 fields in each retina at six weeks after microbeads injection or ten weeks after DEX modeling.

e. Compared with wild-type mice of the same age, the daytime and night IOP of the mice injected with magnetic beads was $7.154\pm 0.5457\text{mmHg}$ higher than that of the WT mice at the 6th week, and the night IOP of the mice was $8.615\pm 0.7634\text{mmHg}$ higher than that of the WT mice. $n=18$, all values were expressed as mean \pm SEM. * $p < 0.05$, ** $p < 0.01$, *** $p < 0.001$, unpaired T-test.

f-g. RGC count result. Compared with WT mice of the same age, the Rbpms+ RGCs of mice injected with microbeads decreased by 16.9%, n=8, and the data were all expressed as mean±SD, *p < 0.05, **p < 0.01, ***p < 0.001.

h. The changes of daytime and night IOP of DEX-induced mice (n=14) in ten weeks compared with WT mice (n=10) of the same age, all values were expressed as mean±SEM. *p < 0.05, **p < 0.01, ***p < 0.001, unpaired T-test.

i-j. The result of RGC counting. Compared with WT (n=10) mice of the same age, the Rbpms+ RGCs of mice weekly injected with DEX (n=14) decreased by 12.2% (n=8), and the data were all expressed as mean±SD, *p < 0.05, **p < 0.01, ***p < 0.001.

2. The authors chose to use a glucocorticoid-induced glaucoma model (by injecting dexamethasone into the eye every 7 days for the duration of the experiment) to carry out single cell transcriptomic analyses. Their sc results show that most of DE genes up-regulated from Y25 relative to controls were mainly related to immune processes. Knowing that dexamethasone and glucocorticoids in general are powerful inhibitors of immune cell function and can modulate the numbers of NK cells, among other immune cells, are they not afraid that this model will mask inflammation processes that otherwise would be much higher “in real life”? In terms of the sc analyses how does this model would compare with the microbead ocular hypertension model? And is there a rationale to why the authors did not use this model instead of the GC one?

Reply: We thank for these useful comments. As raised by the reviewer, glucocorticoids are potent immunosuppressants. We previously assessed various animal models of glaucoma and subsequently conducted gene therapy studies targeting the processes of aqueous humor outflow and aqueous humor generation. To enhance the generalizability of our study, we selected a glucocorticoid-induced ocular hypertension model to simulate chronic open-angle glaucoma, and a magnetic beads model to simulate acute

angle-closure glaucoma. Considering that other disease models are less commonly used than these two models, our choice is appropriate for evaluating whether gene therapy can address multiple types of glaucoma. However, as pointed out by the reviewer, we acknowledge that glucocorticoids possess strong immunosuppressive properties. Nevertheless, our control group samples were obtained from untreated tissues in a glucocorticoid-induced ocular hypertension model, while the experimental group samples were collected after Y25 treatment in the same conditions. Therefore, differential expression genes can accurately reflect the impact of *Rock1* and *Rock2* knockdown on this disease model, ensuring reliable conclusions.

Minor:

3. In addition to RGC data do the authors have any data concerning neuronal loss in the the optic nerve? Or OCT data?

Reply: Thanks for your kindly comment. In this study, we did collect OCT data from each group of model mice 10 weeks after injection of the treatment AAV, with the WT group as the control. However, we did not analyze significant differences in the retinal OCT data. The images are shown in the Fig. S18. This may be because the kinds of model did not cause a large number of RGC deaths in the entire retina during our observation period, but only caused significant RGC deletion in the peripheral retina.

Fig. S18. OCT images of retina in two types of glaucoma models treated with shH10 Y25/Y26.

a. OCT image of retina in each group after monocular magnetic beads modeling and injection of shH10 virus for 10 weeks. Scale bars, 100 μ m.

b. OCT image of retina in each group after DEX modeling and injection of shH10 virus. Scale bars, 100 μ m.

- 1 Basavarajappa, D. *et al.* Signalling pathways and cell death mechanisms in glaucoma: Insights into the molecular pathophysiology. *Molecular Aspects of Medicine* **94**, 101216.
- 2 Hendrix, S. & Nitsch, R. The role of T helper cells in neuroprotection and regeneration. *Journal of Neuroimmunology* **184**, 100-112.
- 3 Bakalash, S., Kipnis, J., Yoles, E. & Schwartz, M. Resistance of retinal ganglion cells to an increase in intraocular pressure is immune-dependent. *Invest Ophthalmol Vis Sci* **43**, 2648-2653 (2002).
- 4 Barış, M. & Tezel, G. Immunomodulation as a neuroprotective strategy for glaucoma treatment. *Current ophthalmology reports* **7**, 160-169 (2019).
- 5 Dressman, D. & Elyaman, W. T Cells: A Growing Universe of Roles in Neurodegenerative Diseases. *The Neuroscientist* **28**, 335-348.
- 6 Wang, L. & Wei, X. T cell-mediated autoimmunity in glaucoma neurodegeneration. *Frontiers in Immunology* **12**, 803485 (2021).
- 7 Sas, A. R. *et al.* A new neutrophil subset promotes CNS neuron survival and axon regeneration. *Nature immunology* **21**, 1496-1505 (2020).
- 8 Zincarelli, C., Soltys, S., Rengo, G. & Rabinowitz, J. E. Analysis of AAV serotypes 1-9 mediated gene expression and tropism in mice after systemic injection. *Mol Ther* **16**, 1073-1080.

REVIEWERS' COMMENTS

Reviewer #1 (Remarks to the Author):

Authors have addressed my comments.

Reviewer #2 (Remarks to the Author):

The authors have adequately addressed most of the reviewer's comments, except for the explanation of how blocking Rho-associated protein kinase (ROCK) inhibitors reduces intraocular pressure (IOP). As noted by the reviewer, this mechanism involves actin depolymerization, which facilitates aqueous humor (AH) outflow and thus lowers IOP. The authors propose that inhibiting ROCK inhibitors leads to increased neutrophil infiltration, which contributes to the reduction of IOP through immune-mediated responses. This interpretation diverges from the commonly understood beneficial effects of ROCK inhibitors, suggesting possible species-specific differences between mice and humans. The reviewer acknowledges the neuroprotective effects of ROCK inhibitors.

Since the description appears in the discussion section, the reviewer accepts it as presented.

Reviewer #3 (Remarks to the Author):

My concerns have been addressed in the revisions

RESPONSE TO REVIEWER COMMENTS

Reviewer #1 (Remarks to the Author):

Authors have addressed my comments.

Reply: We would like to thank you for your professional review work, constructive comments, and valuable suggestions on our manuscript.

Reviewer #2 (Remarks to the Author):

The authors have adequately addressed most of the reviewer's comments, except for the explanation of how blocking Rho-associated protein kinase (ROCK) inhibitors reduces intraocular pressure (IOP). As noted by the reviewer, this mechanism involves actin depolymerization, which facilitates aqueous humor (AH) outflow and thus lowers IOP. The authors propose that inhibiting ROCK inhibitors leads to increased neutrophil infiltration, which contributes to the reduction of IOP through immune-mediated responses. This interpretation diverges from the commonly understood beneficial effects of ROCK inhibitors, suggesting possible species-specific differences between mice and humans. The reviewer acknowledges the neuroprotective effects of ROCK inhibitors.

Since the description appears in the discussion section, the reviewer accepts it as presented.

Reply: We sincerely thank you for your feedback which would help to improve the quality of our manuscript. And thank you for your recognition of the neuroprotective effects of ROCK inhibitors. By RNA-seq analysis, we found that the mice that inhibited *Rock* gene expression did indeed have increased neutrophil infiltration, and explained it, suggesting that IOP might be reduced through an immune-mediated response. This explanation differs from the commonly understood beneficial effects of ROCK inhibitors. As you mentioned, there may be species-specific differences between mice and humans because the model mice we used are difficult to fully simulate human disease conditions. In the future, it is necessary to carry out in-depth research to further explore the internal mechanism of this problem.

Reviewer #3 (Remarks to the Author):

My concerns have been addressed in the revisions

Reply: Thank you for the positive and constructive comments regarding our paper.